# REDUCING HALLUCINATIONS IN GENERATIVE MODELS THROUGH TRUNCATED STATISTICS

## ABSTRACT

Hallucinations—where generative models produce invalid or nonsensical outputs—remain a critical challenge for reliable deployment. We present the first computationally and query-efficient algorithm that provably addresses the hallucination problem by actively querying the model's own invalid outputs. Specifically, we impose a strict constraint on the hallucination rate while maximizing the likelihood of valid target examples via projected stochastic gradient descent. Our method works in very general settings with arbitrary distributions parameterized by sufficiently expressive exponential families. Our approach is enabled by a novel connection to the field of truncated statistics and settles an open problem posed by Hanneke et al. (2018).

## 1 INTRODUCTION

Hallucinations—instances where large language models (LLMs) produce factually incorrect or misleading content—have emerged as a critical challenge to the safe and effective use of AI across numerous domains. In telecommunications, such inaccuracies can lead to miscommunications and flawed decision-making, undermining the reliability of communication services and potentially causing significant operational disruptions (Ji et al., 2023; Liu et al., 2024). Similarly, in autonomous driving, hallucinations can cause LLMs to misinterpret the environment or generate unrealistic predictions, compromising real-time decision-making and overall vehicle safety (Wang, 2024). This issue is particularly concerning as LLM-based self-driving systems aim to bridge the gap between real and virtual environments while maintaining efficiency and minimizing computational overhead. Meanwhile, in medical practice, these same hallucinations pose serious safety risks by generating false or incomplete information, which can jeopardize patient outcomes and erode trust in clinical decision-making tools (Mello & Guha, 2023). As LLMs continue to proliferate and power essential applications, addressing the **hallucination problem** is not only a technical challenge but also a societal imperative to ensure reliability and trust in AI systems.

This work adopts a distribution learning perspective on the hallucination problem. Specifically, we posit that there exists an underlying distribution $p$ which generates data points $x_{1:n} \coloneqq \{x_1, \ldots, x_n\}$ within a space $X$. However, only a subset $V \subset X$ represents valid or accurate data, while the complement $V^c = X \setminus V$ constitutes the hallucination set.

*Our objective is to fit $\theta$ so that $p_\theta$ approximates the true distribution $p$ (i.e., fits the data $x_{1:n}$) while minimizing $p_\theta(V^c)$.*

One straightforward approach to mitigating hallucinations is to form a mixture between our original model and another model $M'$ that always generates a fixed valid output $x^*$, akin to saying "I don't know." By adjusting how often $M'$ is activated, hallucinations can be reduced arbitrarily low. However, while this method ensures that the model never produces invalid outputs, it significantly limits the ability to generate diverse and meaningful responses, ultimately compromising the model's utility. Instead, we seek a balance: the model should provide informative and varied responses while keeping the hallucination rate under control. We formalize this objective as maximizing the likelihood of the observed training data while ensuring that the probability of hallucination remains below a predefined threshold.

More precisely, given a parametric family of distributions $\{p_\theta\}$ over the data space $X$, our objective is to find a distribution $p_{\theta*}$ that maximizes the likelihood of the observed data while ensuring that the

hallucination rate does not exceed $\alpha$:

$$p_{\theta^*} = \arg\min_\theta \sum_{i=1}^n -\log p_\theta(x_i) \ \text{ s.t. } p_\theta[V^c] \leq \alpha. \tag{1}$$

This formulation ensures that the learned distribution remains faithful to the observed data $x_{1:n}$ while appropriately restricting the probability mass assigned to the hallucination set $V^c$, thereby maintaining both reliability and utility.

Passive learning is insufficient for general distributional learning (Hanneke et al., 2018)—and, *inadvertently*, for LLM settings as well. Hanneke et al. (2018) introduce active distribution learning with an invalidity oracle, define Valid Generative Modeling (VGM), and show that *proper* learning can require exponentially many invalidity queries (an information-theoretic barrier), whereas an *improper* learner can be statistically and query efficient. For a relaxed validity constraint, they obtain statistical efficiency but explicitly pose as an open problem whether one can also achieve *computational* efficiency (even with access to an optimization oracle). We resolve this open problem in the affirmative.

**Central question (Hanneke et al. (2018)).** *Does there exist a statistically, query-, and computationally efficient algorithm that fits the data while driving the hallucination rate to an arbitrary target $\alpha$?*

## 1.1 OUR RESULTS

While the problem of maintaining a low hallucination rate is statistically solvable, it is *computationally* intractable in general. For instance, Hanneke et al. (2018) have demonstrated that proper learning in this context is intractable, underscoring the significant challenges in developing computationally efficient models. This intractability highlights the necessity to operate within a family that is expressive enough. Furthermore, even a simple Gaussian parametrization is not expressive enough and the resulting hallucination problem can be NP-hard. Formally:

**Claim 1** (Computational intractability for simple model)**.** *Even when $p_\theta$ is restricted to the Gaussian family $\mathcal{N}(\theta, I)$, it is NP-hard to find a feasible solution to the constrained optimization problem (1) (i.e., to ensure $q[V^c] \leq \alpha$) under this model.*

For a formal proof we defer the reader to Section E. This negative result is disappointing. However, we demonstrate that intractability only arises in models that are insufficiently expressive to *shift* probability mass from the hallucination set $V^c$ to valid regions without unduly sacrificing likelihood on the observed data. To formalize this idea, we introduce the notion of a *powerful model*:

**Definition 2** (Powerful model (informal))**.** *A distribution family $\mathcal{P}$ is powerful at level $\alpha$ if for any $p \in \mathcal{P}$ with $p[V^c] > \alpha$, there exists $p' \in \mathcal{P}$ with $p'[V^c] = \alpha$ such that, for any valid data $x_{1:n} \subseteq V$,*

$$\frac{p'[V^c]}{p[V^c]} \leq \frac{\prod_{i=1}^n p'(x_i)}{\prod_{i=1}^n p(x_i)},$$

*where $V^c$ is the hallucination set.*

Intuitively, the ratio condition in Definition 2 says that if we reduce the hallucination mass by a factor $r = p'[V^c]/p[V^c] < 1$, then the likelihood of the observed valid data $\{x_i\} \subseteq V$ under $p'$ can decrease by at most the same factor: $\prod_i p'(x_i) \geq r \prod_i p(x_i)$. In other words, we can cut hallucinations without the data likelihood collapsing. This property is straightforward for model families that are closed under mixtures—probability mass can be reallocated toward $V$ while keeping likelihood on the observed data from vanishing—and we show exponential families satisfy it (see Section D.2).

With the notion of powerful models established, we are now equipped to state our main theorem, which provides both theoretical and algorithmic guarantees for reducing hallucinations in these expressive families. We state now an informal version of our main theorem Theorem 16.

**Theorem 3** (Generation without Hallucinations (informal))**.** *Let $\{p_\theta\}$ be a powerful exponential family Definition 2 of distributions over a space $X$, with an unknown validity set $V \subseteq X$ with an invalidity oracle $\mathbf{1}_{V^c}(x)$ and a collection of positive examples $x_{1:n}$ drawn from an unknown distribution $p$ over $X$; and a desired maximum hallucination rate $\alpha > 0$.*

*There exists an algorithm that computes a parameter $\hat{\theta} \in \Theta$ such that the loss $\mathcal{L}(\hat{\theta})$ satisfies $\mathcal{L}(\hat{\theta}) \leq \min_{\theta \in \Theta} \{\mathcal{L}(\theta) : \mathbb{P}_{x \sim p_\theta}[x \in V^c] \leq \alpha\} + \varepsilon$, where $\epsilon > 0$ is a small approximation parameter. The probability of generating invalid samples is bounded: $\mathbb{P}_{x \sim p_{\hat{\theta}}}[x \in V^c] \leq \alpha + \varepsilon$. The algorithm makes $\mathrm{poly}(k, \frac{1}{\epsilon}, \log(1/\alpha))$ invalidity queries and runs in $\mathrm{poly}(k, \frac{1}{\epsilon}, \log(1/\alpha))$ time.*

Theorem 3 establishes proper learning for any powerful exponential family. The assumption is mild and purely about expressivity: any parametric family $\{p_\theta\}$ can be embedded into a slightly augmented, powerful family (e.g., by appending a benign auxiliary component that reallocates excess mass from $S$ to a designated valid point). Importantly, this construction only certifies expressivity—it is not the target of our optimizer. Our method trades off data fit and validity and neither relies on nor promotes degenerate "always-valid" mixtures.

## 1.2 OUR TECHNIQUES

Our novelty relies on establishing a bridge between model generation and truncated statistics. To elucidate this connection, we briefly introduce the framework of truncated statistics. In truncated statistics, we consider a parametric family of distributions $\{p_\theta\}$ over a space $X$, where $p_{\theta*}$ represents the true distribution, and $S \subseteq X$ denotes a truncation set. The objective is to estimate the parameter $\theta^*$ based on truncated data samples $x_i \in S$, which are drawn according to the truncated distribution $p_\theta^S(x) = \frac{p_{\theta*}(x)\mathbf{1}_S(x)}{p_{\theta*}(S)}$. To estimate $\theta^*$, truncated statistics leverages $\theta^*$'s universal property as the minimizer of the truncated negative log-likelihood defined by $\mathcal{L}_S(\theta) = -\mathbb{E}_{x \sim p_{\theta*}^S}\left[\log p_\theta^S(x)\right] = -\mathbb{E}_{x \sim p_{\theta*}^S}[\log p_\theta(x)] + \log p_\theta(S) = \mathcal{L}(\theta) + \log p_\theta(S)$, where $\mathcal{L}(\theta)$ is the negative log-likelihood of $p_\theta$. We establish the connection to truncated statistics and model generation by considering a Lagrangian relaxation of the constrained optimization problem (1). Specifically, we replace the validity constraint $p[V^c] \leq \alpha$ as a regularizer term:

$$\min_\theta \big(\mathcal{L}(\theta) + \lambda \cdot p_\theta(V^c)\big), \tag{2}$$

where $\mathcal{L}(\theta)$ is the negative log-likelihood (NLL) of the observed samples $x_{1:n}$ measures the probability mass assigned to the hallucination set $V^c$, and $\lambda \geq 0$ is a hyperparameter balancing these two objectives.

**Truncated Log-Likelihood and the Role of $\lambda$.** A key observation is that when $\lambda = 1$, the objective in (2) corresponds to a *truncated log-likelihood* with the truncation set $V^c$, thereby allowing us to utilize the extensive tools developed in truncated statistics. In exponential families, this objective is convex, enabling efficient optimization via standard gradient-based methods. Generally, in the regime where $\lambda \leq 1$, the objective remains convex, hence ensuring computational efficiency for all $\lambda \in [0, 1]$. Specifically, the instance $\lambda = 0$ corresponds to fitting the model solely based on the observed data without considering the hallucination set. However, when $\lambda > 1$, the objective becomes non-convex, making the optimization landscape more challenging. Intuitively, increasing $\lambda$ places higher importance on penalizing the mass assigned to $V^c$, potentially leading to solutions that reduce hallucinations more aggressively but at the risk of losing convexity. We find that setting $\lambda = 1$ is the ideal choice for our purposes, as it effectively balances the reduction of hallucinations while maintaining computational tractability.

**Optimal Choice of $\lambda$ in Powerful Models.** In powerful model families definition 2 an important insight is that simply setting $\lambda = 1$ already allows the model to *push* all mass off the hallucination set, effectively driving $p_\theta(V^c) \to 0$ as $\lambda \to 1$ from the left. Consequently, there is no advantage in making $\lambda$ larger; doing so would only enter a non-convex regime without additional benefits in reducing hallucinations. This property ensures that powerful models can enjoy both strong hallucination mitigation and computational tractability.

**Algorithmic Overview.** Building on these observations, our algorithm incrementally relaxes the negative log-likelihood $\mathcal{L}(\theta)$ until it finds a feasible solution. Concretely, we fix a sublevel $\overline{L}$ such that $\mathcal{L}(\theta) \leq \overline{L}$, and then solve

$$\min_\theta \big(\mathcal{L}(\theta) + p_\theta(V^c)\big) \quad \text{subject to} \quad \mathcal{L}(\theta) \leq \overline{L}.$$

As $\mathcal{L}(\theta)$ is convex, this constitutes a convex optimization problem. If the resulting solution $\theta$ satisfies $p_\theta(V^c) \leq \alpha$, we halt and perform a binary search on $\overline{L}$ in case of overshooting. Otherwise, if

$p_\theta(V^c) > \alpha$, we increase $\overline{L}$ and repeat. Thus, we reduce the hallucination problem to a continuum of constrained parameter estimation tasks (similar to those appearing in truncated statistics), each corresponding to a hallucination rate $\alpha$.

### 1.3 RELATED WORK

Our work builds upon and extends several lines of research:

**Generative Models and Hallucinations.** Recent studies have identified the tendency of generative models to produce invalid or nonsensical sequences, particularly in complex or constrained environments (Kusner et al., 2017; Janz et al., 2018; Kalai & Vempala, 2024; Wang, 2024; Liu et al., 2024; Mello & Guha, 2023). Our approach advances this field by providing theoretical guarantees.

**Active Learning with Queries.** In supervised learning, the use of queries, such as membership queries, has been pivotal in learning complex hypothesis classes, often achieving near-optimal performance relative to information-theoretical bounds (Angluin, 1987; Jackson, 1997; Hopkins et al., 2020; Diakonikolas et al., 2024). Active learning techniques have primarily been employed to reduce sample complexity by selecting informative examples. However, Hanneke et al. (2018) have demonstrated that active learning is essential for addressing the hallucination problem. This theoretical necessity is further supported by practical training approaches like Reinforcement Learning with Human Feedback (RLHF) in recent advancements, such as (OpenAI et al., 2024). In our work, we build upon this querying paradigm by utilizing invalid samples to reduce hallucinations while maintaining explainability.

**Truncated Statistics.** A pivotal influence on our study is the work of Lee et al. (2023), which builds upon a substantial body of literature in truncated statistics (Daskalakis et al., 2018; 2020b;a; 2021; Ilyas et al., 2020; Kontonis et al., 2019). Truncated statistics addresses the estimation and learning of distributions from selectively censored or truncated data, presenting unique computational and statistical challenges. Similar to Lee et al. (2023), our research operates within the framework of exponential families, adapting and extending tools from truncated statistics to mitigate hallucinations in generative models. By establishing a direct connection between truncated statistics and hallucination reduction, we bridge these fields and demonstrate that it is a natural framework to model the hallucination problem.

In summary, we synthesize generative-model validity, query-based learning, and truncated statistics to obtain provable hallucination reduction in a simplified parametric setting (exponential families with an invalidity oracle). This serves as a necessary precursor for broader LLM applications—any method that claims to reduce hallucinations should at least meet these guarantees in the simplified case—and provides a principled baseline for future LLM-specific work.

## 2 PRELIMINARIES

Here's a compact combined version:

A distribution is in the **exponential family** if $p_\theta(x) = h(x)\exp(\theta^\top T(x) - A(\theta))$, where $h$ is the carrier (weight) function; for a set $S$, write $p_\theta(S) = \mathbb{P}_{x \sim p_\theta}[x \in S]$. The parameter space is $\Theta = \{\theta \in \mathbb{R}^k : A(\theta) < \infty\}$, an open set; under a **minimal representation** (after reparameterizing if needed so that $\theta$ and $T(x)$ are linearly independent), the log-partition $A$ is convex with $\nabla A(\theta) = \mathbb{E}_{x \sim p_\theta}[T(x)]$ and $\nabla^2 A(\theta) = \mathrm{Cov}_{p_\theta}[T(x)]$.

We include the definition of a univariate sub-exponential random variable, which can be used to define a corresponding class for the multivariate case by taking the supremum over the unit sphere. While there are multiple equivalent definitions, the most convenient for our purposes is based on the moment generating function (MGF). Namely,

**Definition 4** (Moment Generating Function). *The* moment generating function *(MGF) of a distribution $D$, denoted by $M_D(t)$, is defined as $M_D(t) = \mathbb{E}_{x \sim D}[e^{tx}]$, provided this expectation exists. The function $M_D(t)$ is defined for all values of $t$ in some interval containing $t = 0$.*

We provide a definition of the sub-exponential distribution, which we will rely on throughout. This definition is similar to the one in (Vershynin, 2010), with a minor adjustment to account for potential

restrictions that arise when scaling moves us outside the natural parameter space $\Theta$ of our model. Note that, aside from some scaling, the definition used here aligns with that in (Lee et al., 2023).

**Definition 5** (Sub-exponential distribution). *Let $x$ be a univariate random variable with zero mean. The random variable $x$ is said to belong to the class $SE(K_1^2, \beta)$ if the following condition holds: $\mathbb{E}[\exp(\lambda x)] \leq \exp(K_1^2 \lambda^2)$, for all $\lambda$ such that $|\lambda| \leq \frac{1}{\beta}$.*

The following proposition provides an equivalent definition for the sub-exponential class, adapted from (Vershynin, 2010) with slight modifications to suit our needs. However, since the proof remains unchanged, we omit it.

**Proposition 6** (Sub-exponential distribution). *Let $x$ be a univariate random variable with zero mean. Fix parameters $K_i > 0$ for $i = 1, 2$, and $\beta > 0$, such that $x \in SE(K_1^2, \beta)$. Then, the following two properties are equivalent. Additionally, the quantities $\max(K_1, \beta)$ and $K_2$, which appear in the properties, differ by a universal constant. (a) The moment generating function (MGF) of $x$ satisfies $\mathbb{E}[\exp(\lambda x)] \leq \exp(K_1^2 \lambda^2)$ for all $\lambda$ such that $|\lambda| \leq \frac{1}{\beta}$. (b) The moments of $x$ satisfy $(\mathbb{E}[|x|^p])^{1/p} \leq K_2 p$ for all $p \geq 1$.*

This class of functions is typically associated with a concentration inequality. The one we use, which can also be found in (Vershynin, 2010), is as follows:

**Fact 7** (Bernstein's Inequality). *Let $x_1, \ldots, x_N$ be independent, identically distributed, mean-zero, sub-exponential random variables belonging in the $SE(K_1^2, \beta)$, and set $z = \max(K_1, \beta)$. Then, for every $t \geq 0$, we have*

$$\mathbb{P}\left(\left|\frac{1}{N}\sum_{i=1}^{N} x_i\right| \geq t\right) \leq 2\exp\left(-cN\min\left(\frac{t^2}{z^2}, \frac{t}{z}\right)\right),$$

*where $c > 0$ is an absolute constant.*

## 2.1 TRUNCATED STATISTICS

**Truncation and $\beta$–truncated oracle.** For a density $p_\theta$ on $\mathbb{R}^d$ and measurable $S \subseteq \mathbb{R}^d$ with $p_\theta(S) > 0$, define the truncation

$$p_\theta^S(x) = \frac{p_\theta(x)\,\mathbf{1}_S(x)}{p_\theta(S)}, \qquad \frac{p_\theta^S(x)}{p_\theta(x)} = \frac{\mathbf{1}_S(x)}{p_\theta(S)}.$$

We call any procedure that, on input $\theta$, returns a sample from the conditional distribution $p_\theta^S$ whenever $p_\theta(S) \geq \beta$ (and may return an arbitrary output otherwise) a $\beta$–*truncated sample oracle*. With a sampler for $p_\theta$ and a membership oracle $M_S(x) = \mathbf{1}_S(x)$, this is implemented by rejection sampling; the expected number of proposals is $p_\theta(S) \geq \beta$ when the condition holds.

**Non-truncated and Truncated NLL** Given some exponential family $p_\theta$ parameterized by $\theta \in \mathbb{R}^k$, the negative log-likelihood (NLL) over a population distribution $q$ is defined as follows: $\mathcal{L}(\theta) = -\mathbb{E}_{x \sim q}[\log p_\theta(x)]$. In what follows we will refer to $\mathcal{L}$ as "non-truncated". Moreover, for a truncated exponential family $p_\theta^S$, such that the support of $q$ is contained in $S$, we define the following "truncated" version of the NLL objective as: $\mathcal{L}_S(\theta) = -\mathbb{E}_{x \sim q}[\log p_\theta^S(x)] = \mathcal{L}(\theta) + \log p_\theta(S)$. The gradient and Hessian of $\mathcal{L}_S(\cdot)$ are given next. $\nabla_\theta \mathcal{L}_S(\theta) = \mathbb{E}_{x \sim p_\theta^S}[T(x)] - \mathbb{E}_{x \sim q}[T(x)]$, $\nabla_\theta^2 \mathcal{L}_S(\theta) = \mathrm{Cov}_{x \sim p_\theta^S}[T(x)]$.

## 3 GENERATION WITHOUT HALLUCINATIONS: AN OPTIMIZATION PROBLEM

In our setup, we work within a parametrized space of distributions $\{p_\theta\}_{\theta \in \Theta}$, defined by $\Theta$, with an associated loss function $\mathcal{L}(\theta)$. A natural choice is for $\mathcal{L}$ to be the negative log-likelihood. The optimization problem of interest is formulated as:

$$\theta_\alpha^* = \arg\min_\theta \mathcal{L}(\theta) \quad \text{s.t. } \mathbb{P}_{x \sim p_\theta}[x \in S] \leq \alpha. \tag{3}$$

To solve this problem, we rely on three resources: (i) an initial sample set $\{x_i\}_{i=1}^n$; (ii) for each $\theta \in \Theta$, sample access to $x \sim p_\theta$; and (iii) an invalidity oracle $M_S(x)$ that returns 1 if $x \in S$ and

0 otherwise. However, the resulting optimization is typically non-convex because the feasibility constraint $\mathbb{P}_{x\sim p_\theta}[x \in S] \leq \alpha$ is itself non-convex—even in simple settings such as a one-dimensional Gaussian.

**Example 8.** *Consider the following scenario: Let $S = (-1, 1)$, and suppose the distribution is a one-dimensional Gaussian, $\mathcal{N}(\mu, 1)$, with $\mu = M$ or $\mu = -M$. As $M \to \infty$, we observe that $\mathbb{P}_{x\sim\mathcal{N}(\pm M,1)}[x \in S] \to 0$, indicating a decreasing probability within the interval $S$. However, $\mathbb{P}_{x\sim\mathcal{N}(0,1)}[x \in S] \approx 0.6827$, suggesting that for small $\alpha < 0.6827$, the parameter space satisfying the constraint becomes disconnected. This example illustrates the non-convex behavior discussed earlier.*

## 4 CONCENTRATION RESULTS FOR TRUNCATED EXPONENTIAL FAMILIES

In order to accomplish that, we have to bound $\mathrm{Cov}_{x\sim p_{\theta*}^S}(T(X))$, and in order to do that we need $\|\mathbb{E}_{x\sim p_\theta^S}[T(x)] - \mu\|_2 \lesssim \log\left(\frac{1}{p_\theta(S)}\right)$. The next lemma will deal with those two latter stepping stones.

**Lemma 9** (Moment preservation after truncation). *Assume that $\mathbb{E}_{x\sim p_\theta}[T(x)] = \mu$, $\mathrm{Cov}_{x\sim p_\theta}(T(x)) \preceq LI$, and $p_\theta(S) > 0$. Then, $\|\mathbb{E}_{x\sim p_\theta^S}[T(x)] - \mu\|_2 \lesssim \log\left(\frac{1}{p_\theta(S)}\right)$. Similarly, we have the following covariance estimate: $\mathrm{Cov}_{x\sim p_\theta^S}(T(x)) \preceq \left(O\left(\log^2\left(\frac{1}{p_\theta(S)}\right)\right) + L\right)I$.*

The previous lemma gives us an estimate of how much the smoothness is affected by the truncation. Hence, tighter bounds guarantee needing fewer samples since the truncated distribution enjoys stronger concentration properties. The analysis is achieved by performing a worst-case analysis.

Next, we see how truncation affects the sub-exponential property that the non-truncated distribution possesses. For the proof we defer the reader to Section C.

**Lemma 10** (Truncated density is sub-exponential). *Let $\mathcal{F}$ be an exponential family with sufficient statistic $T(\cdot)$ such that for any $p_\theta \in \mathcal{F}$, we have $\lambda I \preceq \mathrm{Cov}_{x\sim p_\theta}[T(x)] \preceq LI$. Let $x$ follow the truncated distribution: $p_\theta^S(x) = \frac{p_\theta(x)\mathbf{1}_S(x)}{p_\theta(S)}$, where $p_\theta(x) = h_S(x)\exp(\theta^\top T(x) - A_S(\theta))$ and $p_\theta(S)$ is the normalization constant. Then, the random variable $T(x)$, under this truncated distribution, is sub-exponential, denoted $SE(K^2, \beta)$, with parameters $K^2 = (1 + \log^2(1/\alpha))L$ as $K$ appears in Definition 5, and $\beta$ is the reciprocal of the largest radius $r$ of the ball $B(\theta, r)$, centered at $\theta$, that is contained in $B(\theta, r) \subset \Theta$.*

**Remark:** Corollary 13 shows that we can restrict our attention to $\theta \in \{\theta : \mathcal{L}(\theta) - \mathcal{L}(\theta_0) \leq \log(1/\alpha)\}$. Any such $\theta$ satisfies $\mathbb{P}_{x\sim p_\theta}[x \in S] \geq \mathrm{poly}(\alpha)$, therefore $\log(1/\mathbb{P}_{x\sim p_\theta}[x \in S]) \leq O(\log(1/\alpha))$. Specifically, in Lemma 25, the bound $K^2 = (1 + \log^2(1/\alpha))L$ follows from this observation.

### 4.1 THE FEASIBLE REGION AROUND INITIALIZATION

To ensure feasiblity we will use the following simple observation.

**Observation 11** (Mass along the sub-level sets). *Let $p_\theta$ be some exponential family and define $\theta'$ to be a minimizer of the constrained objective $\{\min_\theta \mathcal{L}_S(\theta) : s.t. \mathcal{L}(\theta) \leq \bar{L}\}$ for some $\bar{L}$. For any $\theta$ such that $\mathcal{L}(\theta) \leq \bar{L}$, it holds $\mathbb{P}_\theta[S] \geq \mathbb{P}_{\theta'}[S]\exp(\mathcal{L}(\theta') - \mathcal{L}(\theta))$.*

For the proof we defer the read to Section C.4. Using Observation 11 the following technical corollary provides the geometry of optimization landscape, for a proof we defer the reader to Section C.5.

**Corollary 12** (Mass monotonicity along sub-level thresholds). *Let $\theta^*$ be the minimizer of the truncated NLL objective $\mathcal{L}_S$ for some exponential family $p_\theta$, constrained on the convex domain $\{\theta : \mathcal{L}(\theta) \leq \mathcal{L}(\theta^*)\}$. Denote by $\alpha = \mathbb{P}_{x\sim p_{\theta*}}[x \in S]$ the mass assigned to the survival set $S$. Define the following constrained optimization problem: $\{\min_\theta \mathcal{L}_S(\theta) : s.t. \mathcal{L}(\theta) \leq \bar{L}\}$ and denote by $\theta_1$ its solution with $\bar{L} = L_1$ and by $\theta_2$ its solution of the same optimization problem with $\bar{L} = L_2$, where $L_2 > L_1$.*

*1. (Monotonicity) The mass assigned to $S$ is decreasing as a function of $\bar{L}$: $\mathbb{P}_{x\sim p_{\theta_1}}(S) \geq \mathbb{P}_{x\sim p_{\theta_2}}[x \in S] \geq \alpha$.*

*2. (Exponential decrease) The mass assigned to $S$ by $\theta_2$ drops exponentially fast, i.e., it holds that* $\mathbb{P}_{x \sim p_{\theta_2}}[x \in S] \leq \mathbb{P}_{x \sim p_{\theta_1}}[x \in S] \, e^{-(L_2 - L_1)}$.

Given the exponential decrease, i.e. Corollary 12 we obtain the following corollary. Which is ensuring that our initialization of PSGD is close in LLN distance, and that all our parameters inside our projection domain give $\mathrm{poly}(1/\alpha)$ to the survival set $S$.

**Corollary 13** (Feasibility of $\theta^0$). *Suppose that $\theta^0$ is such that $\mathbb{P}_{x \sim p_{\theta^0}}[x \in S] \geq \alpha$. Suppose $\theta_\alpha^*$ is the solution of Equation (3). Then, $\mathcal{L}(\theta^*) \leq \mathcal{L}(\theta^0) + \log\left(\frac{1}{\alpha}\right)$. Furthermore, if for any $\theta$ such that $\mathcal{L}(\theta) \leq \mathcal{L}(\theta^0) + \epsilon + \log\left(\frac{1}{\alpha}\right)$, then $p_\theta(S) \geq p_{\theta^*}(S)\alpha e^{-\epsilon}$.*

*Proof.* From Corollary 12 it holds that $p_{\theta^0}(S) > \alpha$. Hence, by the exponential decrease, provided by the same corollary, we find that $\mathcal{L}(\theta^*) \leq \mathcal{L}(\theta^0) + \log\left(\frac{1}{\alpha}\right)$. The second part is an immediate consequence of the Observation 11. $\square$

The following lemma is a smoothness result that specifies how much we need to expand the sub-level sets to avoid overshooting and subsequently reducing the mass by more than a constant factor. For the proof we defer the reader to Section C.6.

**Lemma 14** (Bounded gaps and proximity of a $(\delta, \epsilon)$ minimum). *Let $\theta_L$ the solution $\theta_L = \arg\min_{\theta \in \Theta} \mathcal{L}_S(\theta)$ such that $\mathcal{L}(\theta) \leq L$. Suppose $\overline{\theta}_L$ approximates the loss by $\epsilon$, i.e. $\mathcal{L}_S(\overline{\theta}_L) - \mathcal{L}_S(\theta_L) \leq \epsilon$. Denote $\delta = L - L(\overline{\theta}_L)$, and set $\theta_{L-\delta}$ the solution $\theta_L = \arg\min_{\theta \in \Theta} \mathcal{L}_S(\theta)$ such that $\mathcal{L}(\theta) \leq L - \delta$. Then, the following approximation is true $e^{-\epsilon-\delta}\mathbb{P}_{x \sim p_{\overline{\theta}_L}}[x \in S] \leq \mathbb{P}_{x \sim p_{\theta_L}}[x \in S] \leq e^{-\delta}\mathbb{P}_{x \sim p_{\overline{\theta}_L}}[S]$. Also, $\mathcal{L}_S(\theta_{L-\delta}) - \mathcal{L}_S(\theta_L) \leq \epsilon$. Lastly, for $g = \mathcal{O}\left(\left(\frac{L}{\lambda}\log^2(1/\alpha)\right)^{-1}\right)$, we get $\mathcal{L}_S(\theta_L) - \mathcal{L}_S(\theta_{L+g}) \leq 1$.*

We now show that when consecutive minima are close, the overall progress must be negligible.

**Proposition 15** (Progress of the Truncated Loss along Sublevel Sets). *Suppose $\theta_m^*$ is the minimizer of the optimization problem $\{\min_\theta \mathcal{L}_S(\theta) : s.t. \mathcal{L}(\theta) \leq \overline{L}\}$ for $\overline{L} := L_{\min} + mb$, where $m \in \mathbb{N}, b > 0$. Moreover, let $l$ be the smallest integer, such that $\theta^*$ is the solution of the same constrained minimization problem for $\overline{L} := L_{\min} + lb$. Then, for $m \leq l - 1$, $\mathcal{L}_S(\theta_m^*) - \mathcal{L}_S(\theta_{m+1}^*) \geq \frac{\mathcal{L}_S(\theta_m^*) - \mathcal{L}_S(\theta^*)}{l - m}$ and for all $m$, $\mathcal{L}_S(\theta_m^*) - \mathcal{L}_S(\theta_{m+1}^*) \geq \frac{\mathcal{L}_S(\theta_m^*) - \mathcal{L}_S(\theta^*)}{\log\left(\frac{1}{\alpha}\right)/b}$.*

# 5 LEARNING WITHOUT HALLUCINATIONS

In this section we define the class of distributions named powerful, and show that there the optimization problem Equation (3) can be solved efficiently. A crucial property for the powerful class is that we may define a regularized log-likelihood function for which the approximate minima are pushed to the boundary. We now state the main theorem of this section.

**Theorem 16** (Generation without hallucinations). *Fix $\epsilon > 0$. Let $\{p_\theta\}$ be a powerful exponential family, with sufficient statistic $T : \mathbb{R}^m \to \mathbb{R}^k$ such that for any $p_\theta$, $\lambda I \preceq \mathrm{Cov}_{x \sim p_\theta}[T(x)] \preceq LI$. Fix an unknown subset of $X$, $S$ with oracle access $\mathbf{1}_S$. Suppose $x_1, x_2, \cdots, x_n$ are samples drawn from an unknown distribution, and $\theta_0 = \arg\min_\theta \mathcal{L}(\theta, x_1, x_2, \cdots, x_n)$, such that $\mathbb{P}_{x \sim p_\theta}[x \in S] > \alpha$, and an $\Omega(\alpha)$-truncated sample $M_S$ over $S$. Denote $\theta^*$ as a solution of $\theta^* = \arg\min_\theta \mathcal{L}(\theta)$, with hallucination rate $\alpha$, i.e., such that $\mathbb{P}_{x \sim p_\theta^*}[x \in S] \leq \alpha$. There exists an algorithm that makes $\widetilde{O}\left(\frac{k^2}{\epsilon^2}\log\left(\frac{1}{\alpha}\right)^2\right)$ calls to $M_S$, runs in time $\mathrm{poly}(k, 1/\epsilon, \log(\frac{1}{\alpha}))$ and computes an estimate $\hat{\theta}$, such that $\mathrm{KL}(p_{\theta^*}^S \| p_{\hat{\theta}}^S) \leq \epsilon$ and $\mathbb{P}_{x \sim p_{\hat{\theta}}}[x \in S] \leq (1 + \epsilon)\alpha$ with probability at least 99%.*

To minimize the number of queries to the oracle, our approach cautiously progresses towards the optimal parameter $\theta_\alpha^*$, see Equation (3). Specifically, we incrementally explore increasingly larger convex sets, ensuring each set assigns sufficient probability mass to the set $S$. To this end, for a given threshold $\overline{L}$, we formulate and solve the following constrained optimization problem:

$$\theta^* = \arg\min_\theta \mathcal{L}_S(\theta) \quad \text{s.t. } \mathcal{L}(\theta) \leq \overline{L} \tag{4}$$

**Roadmap of the proof** We describe the main ideas for proving Theorem 16. *First*, we note that the initial point $\theta_0$ that minimizes the observed samples is close to $\theta_\alpha^*$ as in Equation (3). More precisely $\theta^* \in \{\theta : \mathcal{L}(\theta) \leq \mathcal{L}(\theta_0) + O(\log \frac{1}{\alpha})\}$. To this end, we quantify the rate at which the mass assigned by the minima to $S$ decreases along the sublevel sets. For this a crucial both observation 11 to see how much far we need to search and lemma 14 to make sure we do not overshoot too much. *Second*, we implement projected gradient descent along the sublevel sets. This process requires identifying when we are within the sublevel set where the true minimum lies.

The Algorithm 1 generates the desired output, $\hat{\theta}$, while incorporating all subsequent algorithms. The algorithm terminates at line 4 when the successive outputs for Equation (4), corresponding to the sublevel sets $L_1$ and $L_2$ (with $L_2 - L_1 = g$), differ by no more than $\epsilon''$. The parameter $\epsilon''$ is chosen to be small enough such that, at termination, we are within $\epsilon$ of the minimizer of $\mathcal{L}_S(\theta)$ so that we are close to boundary. This comparison is justified by Proposition 15, which compares the successive distances to the distance from the global minimum.

---

**Algorithm 1** Algorithm for reducing the hallucination rate to $\alpha$

---

1: **for** $\bar{L} \in \left\{ L_{\min} + g, \, L_{\min} + 2g, \, \ldots, \, L_{\min} + \left( \lfloor \log(1/\alpha)/g \rfloor + 1 \right)g \right\}$ **do**
2:      Execute PSGD (Algorithm 2) on sublevel $\bar{L}$ and obtain $\theta_{\bar{L}}$
3:      Compute $\mathcal{L}_S(\theta_{\bar{L}})$
4:      **if** $\mathbb{P}_{x \sim p_{\theta_{\bar{L}}}}[x \in S] \, \exp\left( \bar{L} - \mathcal{L}(\theta_{\bar{L}}) \right) \leq \alpha$ **then**
5:          **return** `binary-search`$\left( \mathcal{L}(\theta_{\bar{L}}) \right)$ (Algorithm 4)
6:      **end if**
7:      **if** $\left| \mathcal{L}_S(\theta_{\bar{L}-g}) - \mathcal{L}_S(\theta_{\bar{L}}) \right| \leq \varepsilon''$ **then**
8:          **return** $\theta_{\bar{L}}$
9:      **end if**
10: **end for**

---

## 5.1 POWERFUL MODEL

Here, we develop the framework of powerful models.

**Definition 17** (Powerful model). *Let $F$ be a family of probability density functions on a space $X$, and let $S \subset X$ be a measurable subset. Suppose we have observed data $\{x_i\}_{i=1}^n \subset X$. The family $F$ is called $\alpha$-**powerful with respect to** $S$ if, for all $p \in F$ such that $p(S) > \alpha$, there exists $p' \in F$ satisfying $p'(S) = \alpha$ and: $\frac{p'(S)}{p(S)} \leq \exp\left( \frac{1}{n} \sum_{i=1}^n \left( \ln p'(x_i) - \ln p(x_i) \right) \right)$. Should the family $F$ be powerful for all $\alpha$ we simply call $F$ powerful.*

**Definition 18** ($\alpha$-Closed Under Mixtures). *Let $\mathcal{F}$ be a space of families of probability distributions over a domain $X$, and let $S \subset X$ be a measurable subset. Extend $X$ by adding a point $x^* \notin X$, forming $X' = X \cup \{x^*\}$. For a family $F \in \mathcal{F}$ and any $p \in F$ with $p(S) > \alpha$, define the extended distribution $p_m$ on $X'$ as:*

$$p_m(x) = \begin{cases} p(x), & \text{if } x \in X \setminus S, \\ p(x) \cdot \frac{\alpha}{p(S)}, & \text{if } x \in S, \\ p(S) - \alpha, & \text{if } x = x^*. \end{cases}$$

*We say that $\mathcal{F}$ is $\alpha$-**closed under mixtures** with respect to $S$ if there exists a family $F' \in \mathcal{F}$ such that $p_m, p \in F'$ for all $p \in F$ with $p(S) > \alpha$, and the extension $p(x^*) = 0$ holds for distributions in $F$.*

The next lemma shows that if we can output something valid with increased probability, then the model must be powerful. For the proof of this lemma see Section D.1.

**Lemma 19** (Closed Under Mixtures Implies Powerful). *If a family $F$ is $\alpha$-closed under mixtures with respect to $S$, then $F$ is $\alpha$-powerful with respect to $S$.*

An exponential family can be thought as closed under mixtures. See Section D.2 for the proof.

**Corollary 20** (Exponential Family is $\alpha$-Closed Under Mixtures). *Let $\{p_\theta\}$ be an exponential family of probability density functions on a space $X$, with sufficient statistic $T(x)$ and base measure $h(x)$. Then, for any $\alpha \in (0, 1]$, the exponential family is $\alpha$-closed under mixtures with respect to any measurable subset $S \subset X$.*

Putting all the pieces together we conclude this section with the following corollary.

**Corollary 21** (Exponential Families are $\alpha$-Powerful). *Since the exponential family is $\alpha$-closed under mixtures (Corollary 20), and closed under mixtures implies powerful (Lemma 19, it follows that the exponential family is $\alpha$-powerful with respect to any measurable subset $S \subset X$.*

## 5.2 APPROXIMATE MINIMA LIE ON THE BOUNDARY

Here, we develop the main technical tools necessary for our approach, demonstrating that in powerful models, there always exists a direction in which the NLL decreases, along with a corresponding proxy objective that leverages this.

**Proposition 22** (A direction in which $\mathcal{L}_S$ is decreasing). *Suppose that $\{p_\theta\}_{\theta \in \Theta}$ forms an exponential family that is powerful (see Definition 17) with respect to a set $S \subset X$, taking values in $X$. Let $A \subset \Theta$ be such that $\max_{\theta, \theta' \in A} \|\theta - \theta'\|_2 \leq M$, and let $\alpha < \min_{\theta \in A} \mathbb{P}_{x \sim p_\theta}[x \in S]$. Then, there exists a unit vector $u$ such that for all $\theta \in A$, we have $u^T \nabla \mathcal{L}_S(\theta) \leq 0$. Furthermore, if we sample a random unit vector $v$, then with probability at least $\Theta(1)$, we have that $u^T v / \|v\|_2 \geq a$, where $a = \Theta(k^{-1/2})$, where $k$ the dimension of $\theta$.*

*Proof.* Fix $\epsilon > 0$ and observe that, since $A$ is a bounded set, there exists a point $\theta_\epsilon$ such that for all $\theta \in A$, $\mathcal{L}_S(\theta_\epsilon) < \mathcal{L}_S(\theta)$, and $\frac{(\theta_\epsilon - \theta_1)^T (\theta_\epsilon - \theta_2)}{\|\theta_\epsilon - \theta_1\|_2 \|\theta_\epsilon - \theta_2\|_2} \geq 1 - \epsilon$, for all $\theta_1, \theta_2 \in A$. On the other hand, if we take $\epsilon \to 0$, then, by the compactness of the unit sphere, there exists a unit direction $u$ such that $\mathcal{L}_S(\theta + tu)$ is decreasing as function of t, for all fixed $\theta \in A$ and $t \geq 0$. For the final part notice that $\|v\| = \sqrt{k}$ for large $k$ by the law of large numbers. So, $u^T v / \|v\|_2 \sim \mathcal{N}(0, \frac{1}{k})$. $\qquad\square$

Next, we define a regularized version of $\mathcal{L}_S(\theta)$ that we will later see that it has the property that its minima lie close to the boundary.

**Definition 23** ($\epsilon$-Regularized loss function). *Suppose that $\{p_\theta\}_{\theta \in \Theta}$ forms an exponential family taking values in $X$, and let $S \subset X$. Define $\mathcal{L}_S^{\epsilon, v}(\theta) = \mathcal{L}_S(\theta) + \epsilon v^T(\theta - \theta_0)$, where $\mathcal{L}_S$ is the truncated negative log-likelihood over the truncated density $p_S$, and $u$ is a unit vector acting on the parameter space $\Theta$.*

The next proposition is very crucial to our analysis, it shows that the regularized loss cannot admit approximate minima as interior point. For a proof see Section D.3.

**Proposition 24** (Minima lie near the boundary). *Suppose that $\{p_\theta\}_{\theta \in \Theta}$ forms an exponential family that is powerful (see Definition 17) with respect to a set $S \subset X$, taking values in $X$. Let $v$ be a vector satisfying $u^T v \geq a$, where $u$ is the direction of decrease as defined in proposition 22. Define $A \subset \Theta$ as the domain $A = \{\theta \in \Theta : \mathcal{L}(\theta) - \mathcal{L}(\theta_0) \leq M\}$, and set $\theta_M^* = \arg\min_{\theta \in A} \mathcal{L}(\theta)$. Then, an $\tilde{\epsilon}$-minimizer $\theta_M$ of $\mathcal{L}_S^{\epsilon', v}$ over $A$, where $\tilde{\epsilon} = O(\epsilon' \epsilon a \log(1/p_{\theta_M^*}(S)))$ satisfies $M + \mathcal{L}(\theta_0) - \epsilon \leq \mathcal{L}(\theta_M)$, where $\mathcal{L}_S^{\epsilon', v}$ is the $\epsilon'$-regularized loss function (see definition 23), and $\mathcal{L}(\theta)$ is the negative log-likelihood of the exponential family.*

## 5.3 SKETCH OF THEOREM 16

By Proposition 24 we pass to the truncated loss $\mathcal{L}_S$, preserving "minima on the boundary" up to a linear penalty in $\tilde{\epsilon}$; on each sublevel $\bar{L}$ we run PSGD and, via Theorem 26 plus amplification and a Hoeffding estimate of $\mathbb{P}_{x \sim p_\theta}[x \in S]$, obtain with high probability a candidate $\theta_{\bar{L}}$ that is $O(\epsilon)$-optimal for that sublevel. The outer routine Algorithm 1 then either (i) overshoots the target mass $\alpha$, in which case Lemma 14 shows the PSGD iterate output $\hat{\theta}$ and the true sublevel minimizer have close mass and $\mathcal{L}$ (hence proximal $\mathcal{L}_S$), so a binary search between the last two visited sublevels returns $\log\left(\mathbb{P}_{x \sim p_{\hat{\theta}}}[x \in S]\right) = \alpha \pm O(\epsilon)$ while keeping $\mathcal{L}(\hat{\theta})$ within $O(\epsilon)$ of $\mathcal{L}(\theta_{\alpha^*})$ thus also keeping $\mathcal{L}_S(\hat{\theta})$ within $O(\epsilon)$ of $\mathcal{L}_S(\theta_{\alpha^*})$; or (ii) does not overshoot, in which case small successive sublevel gaps trigger termination and Proposition 15 implies any further descent could improve $\mathcal{L}_S$ by at most $O(\epsilon)$, so the current iterate is already $O(\epsilon)$-optimal. Using boundary inheritance again to translate $\mathcal{L}(\hat{\theta})$ proximity to $\mathcal{L}(\theta_{\alpha^*})$ (and thus mass), we conclude that $\hat{\theta}$ satisfies $\mathcal{L}_S(\hat{\theta}) - \mathcal{L}_S(\theta_{\alpha^*}) \leq O(\epsilon)$ along with $\log\left(\mathbb{P}_{x \sim p_{\hat{\theta}}}[x \in S]\right) = \alpha \pm O(\epsilon)$ which completes Theorem 16.

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

## A  PSGD CONVERGENCE

We show that the sufficient statistics of the truncated densities are sub-exponential, see Section C.

**Lemma 25** (Sub-exponential Property for the Truncated Distribution). *Let $x$ follow the truncated distribution: $p_\theta^S(x) = \frac{p_\theta(x)\mathbf{1}_S(x)}{p_\theta(S)}$, where $p_\theta(x) = h(x)\exp(\theta^\top T(x) - A(\theta))$ and $\mathbb{P}_{x\sim p_\theta}[S]$ is the normalization constant. Then the random variable $T(x)$ under this truncated distribution is subexponential. More specifically, it belongs to the family $SE(K,\beta)$ where $K^2 = L + \log^2\left(\frac{1}{\alpha}\right)$, $\beta^{-1} = \inf_{\|u\|=1}\sup\{\gamma : \gamma u + \theta \in \Theta\}$.*

The result for PSGD we will use is the following. Its proof is standard, see, .e.g, (Shalev-Shwartz & Ben-David, 2014) or Section C for a proof.

**Theorem 26** (Convergence of Projected Stochastic Gradient Descent). *Consider the Projected Stochastic Gradient Descent (PSGD) algorithm for minimizing a convex function $f$ over a convex set $\mathcal{K}$. Assume the following:*

1. ***Lipschitz Continuous Gradients**: There exists a constant $M > 0$ such that for all $x$ and $y$, $\|\nabla f(x) - \nabla f(y)\| \le M\|x - y\|$.*

2. ***Bounded Variance**: The variance of the stochastic gradients is bounded, i.e., there exists a constant $b^2$ such that $\mathbb{E}[\|g^{(t)}\|^2] \le b^2$, where $g^{(t)} = \nabla f(x_t)$.*

*Then, for $\tilde{w} = \frac{1}{T}\sum_{i=1}^T w^{(i)}$ where the $\{w^{(i)}\}$ are generated by the PSGD algorithm with step size $h_t = \frac{1}{Mt}$, it holds $\mathbb{E}[f(\tilde{w})] - f(w^*) \le \frac{\|w^{(0)} - w^*\|^2}{2MT} + \frac{b^2\pi^2}{12M^2T}$.*

To apply Theorem 26 we first show that our unbiased gradients have bounded second moment. The proof can be found in Section C.

**Lemma 27** (Bounded variances of stochastic gradients ). *Let $T(z) : \mathbb{R}^d \mapsto \mathbb{R}^k$ be the sufficient statistic of a truncated exponential family $p_{\theta^*}^S$. We have that $v = T(z) - T(x)$, where $z \sim p_\theta^S$ and $x \sim p_{\theta^*}^S$ is an unbiased estimator of $\nabla \mathcal{L}_S(\theta)$. Moreover, $v$ has bounded second moment, namely, $\mathbb{E}[\|v\|^2] = \mathbb{E}_{z\sim p_\theta^S}\mathbb{E}_{x\sim p_{\theta^*}^S}\left[\|T(z) - T(x)\|^2\right] \lesssim kL\log^6(1/\alpha)$.*

## B  ALGORITHMIC IMPLEMENTATION DETAILS

In this appendix, we provide detailed descriptions of algorithms used in Theorem 16. These algorithms include the Projected Stochastic Gradient Descent (PSGD) algorithm for optimizing the loss function within a sublevel set, the sampling procedure for computing stochastic gradients via rejection sampling, and the Binary Search procedure for finding the optimal parameter $\theta$ that satisfies the hallucination constraint when we overshoot and need to scale back.

---

**Algorithm 2** Projected SGD with Truncated Samples

---

**Require:** Sublevel threshold $\bar{L}$; positive samples $\{x_i\}_{i=1}^n$ from $p_{\theta^*}^S$; initial parameter $\theta_0 \leftarrow \frac{1}{n}\sum_{i=1}^n T(x_i)$; step-size parameter $L$
1: **for** $t = 1$ **to** $N$ **do**
2:      Sample $x^{(t)}$ from the positive examples
3:      $h_t \leftarrow \frac{1}{L\,t}$
4:      $g^{(t)} \leftarrow \textsc{SampleGradient}(\theta_t, x^{(t)})$
5:      $\theta_{t+1} \leftarrow \theta_t - h_t\, g^{(t)}$
6:      Project $\theta_{t+1}$ onto $\{\,\theta : \mathcal{L}(\theta) \leq \bar{L}\,\}$
7: **end for**
8: **return** $\theta_N$

---

**Algorithm 3** Sampling Procedure for Stochastic Gradient

---

**Require:** Current parameter $\theta$, data point $x$
1: **while** true **do**
2:      Sample $z \sim p_\theta$
3:      **if** $M_S(z) = 1$ **then**                      $\triangleright$ $M_S$ is the invalidity oracle indicating $z \in S$
4:          **return** $T(z) - T(x)$
5:      **end if**
6: **end while**

---

### B.1 Projected Stochastic Gradient Descent (PSGD)

The PSGD (Algorithm 2) is utilized to optimize the loss function $\mathcal{L}(\theta)$ while ensuring that the parameter updates remain within a specified sublevel set defined by $\bar{L}$. At each iteration, the algorithm computes a stochastic gradient using samples from the data distribution and adjusts the parameter $\theta$ accordingly, followed by projection back onto the feasible set.

### B.2 Sampling Procedure for Gradient Computation

The computation of the stochastic gradient requires sampling from the truncated distribution $p_\theta^S$. The sampling procedure (see Algorithm 3) employs rejection sampling using the invalidity oracle $M_S(z)$ to ensure that only valid samples from $S$ are used in the gradient computation.

### B.3 Binary Search for Optimal $\theta$

In some cases, during the iterative optimization, the PSGD algorithm may overshoot the optimal parameter that satisfies the hallucination constraint, resulting in a hallucination rate below the desired maximum $\alpha$. To adjust for this, we employ a Binary Search algorithm (Algorithm 4) to efficiently find the parameter $\theta$ that precisely meets the hallucination rate requirement. This procedure scales back the parameter to ensure compliance with the constraint.

The binary search is needed when the parameter moves beyond the optimal point when we increase the sublevel set , resulting in a hallucination rate that is too low (i.e., the model is overly conservative). By performing binary search, we can efficiently find a parameter that satisfies the hallucination constraint $\epsilon$-close.

## C Omitted Proofs

### C.1 The proof of Proposition 15

*Proof.* Suppose $u = \frac{\theta^* - \theta_m^*}{\|\theta^* - \theta_m^*\|_2}$. Then, the functions

$$g(t) = \mathcal{L}_S(\theta_m^* + tu), \quad w(t) = \mathcal{L}(\theta_m^* + tu),$$

are convex. Furthermore, the functions $g$ and $w$ are decreasing and increasing, respectively, for $t \in [0, \|\theta^* - \theta_m^*\|_2]$. Denote by $t_k$ the points such that $w(t_k) = L' + mb + kb$, where $k =$

---

**Algorithm 4** Binary Search for Optimal $\theta$

---

**Require:** Left bound Left $= L$, right bound Right $= g$, initial params $\theta_0$
 1: **for** $i = 1$ to $\lceil \log(1/\epsilon) \rceil + 1$ **do**
 2:     $\bar{L} \leftarrow (\text{Left} + \text{Right})/2$
 3:     Execute PSGD (Algorithm 2) on sublevel $\bar{L}$ to obtain $\theta_i$
 4:     Estimate $\mathbb{P}_{x \sim p_{\theta_i}}[x \in S]$
 5:     **if** $\mathbb{P}_{x \sim p_{\theta_i}}[x \in S] < \alpha$ **then**
 6:         Right $\leftarrow \bar{L}$
 7:     **else**
 8:         Left $\leftarrow \bar{L}$
 9:     **end if**
10: **end for**
11: **return** $\theta_{\lceil \log(1/\epsilon) \rceil + 1}$

---

$0, \ldots, l - k - 1$ and $t_{l-m} = \|\theta^* - \theta^*_m\|_2$. Since the function $w$ is convex and increasing, the length of the line segments $[t_{k-1}, t_k]$ is decreasing. Also, since $g$ is convex and decreasing,

$$g(t_k) - g(t_{k-1}) \leq g(t_{k-1}) - g(t_{k-2}), \quad \text{for } m = 1, \ldots, l.$$

Consequently,

$$
\begin{aligned}
\mathcal{L}_S(\theta^*_m) - \mathcal{L}_S(\theta^*) &= g(0) - g(\|\theta^* - \theta^*_m\|_2) \\
&= \sum_{i=1}^{l-m} (g(t_{i-1}) - g(t_i)) \\
&\leq (l - m)(g(t_0) - g(t_1)) \\
&\leq (l - m)\left(\mathcal{L}_S(\theta^*_m) - \mathcal{L}_S(\theta^*_{m+1})\right).
\end{aligned}
$$

Hence, we divide by $l - m$ and obtain the first part. Also, since $l - m \leq \log\left(\frac{1}{\alpha}\right)$, we conclude.

$\square$

### C.2 THE PROOF OF LEMMA 25

The truncated exponential family distribution can be written as:

$$p_\theta^S(x) = h_S(x) \exp(\theta^\top T(x) - A_S(\theta)),$$

where:

- $h_S(x) = h(x) 1_S(x)$ is the modified base measure, zero outside the set $S$,
- $A_S(\theta) = A(\theta) + \log p_\theta(S)$ is the modified log-partition function reflecting truncation.

Define the truncated log-partition function:

$$A_S(\theta) = \log \int_S h(x) \exp(\theta^\top T(x)) \, dx.$$

The expected value and covariance under truncation are given by:

$$\mu_S = \nabla A_S(\theta),$$

$$\text{Cov}_{t \sim p_\theta^S}(T(x)) = \nabla^2 A_S(\theta).$$

Consider the moment generating function (MGF): To simplify the calculations deal with the push-forward measure induced by the mapping $x \to T(x)$ and we denote the corresponding density as $p_\theta^S(t) = h_S(t) \exp(\theta t - A_S(\theta))$.

$$E_{t \sim p_\theta^S}[e^{\gamma u^\top (t - \mu_S)}] = e^{-\gamma u^\top \mu_S} \frac{Z_S(\gamma u + \theta)}{Z_S(\theta)},$$

and to establish subexponentiality, we need to establish an inequality of the form:

$$\frac{Z_S(\gamma u + \theta)}{Z_S(\theta)} \cdot e^{-\gamma u^\top \mu_S} \leq e^{\gamma^2 K^2}$$

for $\gamma$ such that both $\theta + \gamma u$ and $\theta + \gamma u$ belong in our parameter space. Using the $\left(L + \log^2\left(\frac{1}{\alpha}\right)\right)$-smooth property of $A_S(\theta)$:

$$A_S(\gamma u + \theta) - A_S(\theta) \leq \gamma u^\top \mu_S + \left(L + \log^2\left(\frac{1}{\alpha}\right)\right)\gamma^2,$$

completing the proof that $T(x)$ under the truncated distribution retains sub-exponential behavior.

$$E_{t \sim p_\theta^S}[e^{\gamma u^\top (t - \mu_S)}] = e^{-\gamma u^\top \mu_S} \frac{Z_S(\gamma u + \theta)}{Z_S(\theta)} \leq \exp\left(\gamma^2\left(L + \log^2\left(\frac{1}{\alpha}\right)\right)\right)$$

Therefore, $T(x)$ is $SE(K^2, \beta)$ with

$$K^2 = \left(L + \log^2\left(\frac{1}{\alpha}\right)\right), \quad \beta^{-1} = \min\left(\sup\{\gamma : \gamma u + \theta \in \Theta\}, \sup\{\gamma : -\gamma u + \theta \in \Theta\}\right).$$

### C.3 THE PROOF OF LEMMA 9

First, for convenience, suppose that $p_\theta(S) = \alpha$. To establish the bound, we will show that for any vector $\vec{u} \in \mathbb{R}^d$, the following holds:

$$\left|\mathbb{E}_{x \sim p_\theta^S}[u^T T(x)] - \mathbb{E}_{x \sim p_\theta}[u^T T(x)]\right| \leq O\left(\log\left(\frac{1}{\alpha}\right)\right).$$

Indeed, for all vectors $w$ and all norms $\|\cdot\|$, there is a linear functional $f$ such that $f(w) = \|w\|$. Since every linear functional is of the form $w \to u^T \cdot w$, we can be sure that, after choosing the proper $u$, we can get a bound for any norm that may be of use.

Define $C_R = \{x : u^T T(x) > R\}$. We change the measure in the first expectation, so we can compare them more readily, namely,

$$\left|\mathbb{E}_{x \sim p_\theta^S}[u^T T(x)] - \mathbb{E}_{x \sim p_\theta}[u^T T(x)]\right|$$

$$= \left|\mathbb{E}_{x \sim p_\theta}\left[\frac{p^S}{p} u^T T(x)\right] - \mathbb{E}_{x \sim p_\theta}\left[u^T T(x)\right]\right|$$

$$= \left|\mathbb{E}_{x \sim p_\theta}\left[\left(\frac{\mathbf{1}_S - \alpha}{\alpha}\right) u^T T(x)\right]\right|$$

$$= \left|\mathbb{E}_{x \sim p_\theta}\left[\left(\frac{\mathbf{1}_S - \alpha}{\alpha}\right) u^T T(x)\mathbf{1}_{C_R}\right] + \mathbb{E}_{x \sim p_\theta}\left[\left(\frac{\mathbf{1}_S - \alpha}{\alpha}\right) u^T T(x)\mathbf{1}_{C_R^c}\right]\right|$$

$$\leq \left|\mathbb{E}_{x \sim p_\theta}\right|\left(\frac{\mathbf{1}_S - \alpha}{\alpha}\right) u^T T(x)\mathbf{1}_{C_R}\right| + \left|\mathbb{E}_{x \sim p_\theta}\left[\left(\frac{\mathbf{1}_S - \alpha}{\alpha}\right) R\right]\right|$$

$$\leq \left|\mathbb{E}_{x \sim p_\theta}\left[\left(\frac{\mathbf{1}_S - \alpha}{\alpha}\right) u^T T(x)\mathbf{1}_{C_R}\right]\right| + \left|\mathbb{E}_{x \sim p_\theta}\left[\left(\frac{\mathbf{1}_S - \alpha}{\alpha}\right) R\right]\right|$$

$$\leq \frac{1}{\alpha}\mathbb{E}_{x \sim p_\theta}\left[u^T T(x)\mathbf{1}_{C_R}\right] + \mathbb{E}_{x \sim p_\theta}\left[\left|\frac{\mathbf{1}_S - \alpha}{\alpha}\right| R\right]$$

$$\leq \frac{1}{\alpha}\mathbb{E}_{x \sim p_\theta}\left[u^T T(x)\mathbf{1}_{C_R}\right] + R$$

It remains to work on the term $\mathbb{E}_{x \sim p_\theta}\left[u^T T(x)\mathbf{1}_{C_R}\right]$:

$$\mathbb{E}_{x \sim p_\theta}\left[u^T T(x)\mathbf{1}_{C_R}\right] = \mathbb{E}_{x \sim p_\theta}\left[u^T T(x)|C_R\right] p_\theta(C_R)$$

$$= \mathbb{E}_{x \sim p_\theta} \left[ \log(\exp u^T T(x) | C_R) \right] p_\theta(C_R)$$

$$= \log \left( \mathbb{E}_{x \sim p_\theta} [\exp u^T T(x) | C_R] \right) p_\theta(C_R)$$

$$\leq \log \left( \mathbb{E}_{x \sim p_\theta} [\exp u^T T(x) \mathbf{1}_{C_R}]) \frac{1}{p_\theta(C_R)} \right) p_\theta(C_R)$$

$$\leq \log \left( \mathbb{E}_{x \sim p_\theta} [\exp u^T T(x) \mathbf{1}_{C_R}]) \frac{1}{p_\theta} \right) p_\theta(C_R)$$

$$\leq \log \left( \mathbb{E}_{x \sim p_\theta} [\exp u^T T(x) \mathbf{1}_{C_R}]) \right) p_\theta(C_R) - \log \left( p_\theta(C_R) \right) p_\theta(C_R)$$

We switch our focus to the term $\log \left( \mathbb{E}_{x \sim p_\theta} [\exp u^T T(x) \mathbf{1}_{C_R}] \right)$

$$\log \left( \mathbb{E}_{x \sim p_\theta} [\exp u^T T(x) \mathbf{1}_{C_R}]) \right) \leq \frac{1}{2} \log \left( \mathbb{E}_{x \sim p_\theta} \left[ \exp(2u^T T(x)) \right] \mathbb{E}_{x \sim p_\theta} [\mathbf{1}_{C_R}] \right)$$

$$\leq \frac{1}{2} \log \left( \mathbb{E}_{x \sim p_\theta} \left[ \exp(2u^T T(x)) \right] \right) \cdot \log \left( \mathbb{E}_{x \sim p_\theta} [\mathbf{1}_{C_R}]) \right)$$

$$\leq \frac{1}{2} \log \left( \mathbb{E}_{x \sim p_\theta} \left[ \exp(2u^T T(x)) \right] \right) \cdot \log \left( p_\theta(C_R) \right)$$

$$\leq \frac{1}{2} \log \left( \exp \left( 4L + 2u^T \mathbb{E}_{x \sim p_\theta} [T(x]) \right) \right) \cdot \log \left( p_\theta(C_R) \right)$$

$$\leq \frac{1}{2} \left( 4L + 2u^T \mathbb{E}_{x \sim p_\theta} [T(x)] \right) \cdot \log \left( p_\theta(C_R) \right)$$

For convenience, we set $p_\theta(C_R) = c(R)$, and by putting everything together, we obtain

$$\left| \mathbb{E}_{x \sim p_\theta^S} [u^T T(x)] - \mathbb{E}_{x \sim p_\theta} [u^T T(x)] \right| \leq \frac{1}{2\alpha} \left( 4L + 2u^T \mathbb{E}_{p_\theta} [T(x)] \right) \cdot \log \left( c(R) \right) c(R)$$

$$- \frac{1}{\alpha} \log(c(R)) \cdot c(R) + R$$

Since $u^T T(x)$ is subexponential, we have that $p_\theta(C_R) \leq C \exp(-cLR)$. Where the constant $C = O \left( \log \mathbb{E}_{x \sim p_\theta} [p(x)] \right)$, which is paid since $p(x)$ is not centered. Substituting back into the original inequality, we get a bound of the form

$$O \left( \frac{1}{\alpha} \exp \left( -cL \cdot R \right) \right) + R.$$

We used the big $O$ notation to suppress any constants, which we may eliminate by paying only $\log(\text{constant})$. Therefore, since the above holds true for all $R$, we may minimize it or choose an $R$ that is satisfactory for our purposes. We may choose $R = O(\log(1/\alpha))$. Therefore,

$$O \left( \frac{1}{\alpha} \exp \left( -cL \cdot R \right) \right) + R = O \left( \log \left( \frac{1}{\alpha} \right) \right),$$

so we conclude the first part.

For the second part we use a similar analysis. To establish the bound, we will bound the corresponding quadratic form of the matrix $\text{Cov}_{x \sim p_\theta^S}(T(x))$. It suffices to bound

$$\left| \mathbb{E}_{x \sim p_\theta^S} \left[ u^T T(x) \cdot T^T(x) u \right] - \mathbb{E}_{x \sim p_\theta} \left[ u^T T(x) \cdot T^T(x) u \right] \right|.$$

To simplify the expression we set $u^T \cdot T(x) = p(x)$ where $p(x)$ is a polynomial of degree $k$.

$$\left| \mathbb{E}_{x \sim p_\theta^S} \left[ u^T T(x) \cdot T^T(x) u \right] - \mathbb{E}_{x \sim p_\theta} \left[ u^T T(x) \cdot T^T(x) u \right] \right|$$

$$= \left| \mathbb{E}_{x \sim p_\theta^S} \left[ p^2 \right] - \mathbb{E}_{x \sim p_\theta} \left[ p^2(x) \right] \right|$$

$$= \left| \mathbb{E}_{x \sim p_\theta} \left[ \left( \frac{\mathbf{1}_S - \alpha}{\alpha} \right) p^2(x) \right] \right|$$

$$\leq \left| \mathbb{E}_{x \sim p_\theta} \left[ \left( \frac{\mathbf{1}_S - \alpha}{\alpha} \right) p^2(x) \mathbf{1}_{C_R} \right] + \mathbb{E}_{x \sim p_\theta} \left[ \left( \frac{\mathbf{1}_S - \alpha}{\alpha} \right) R^2 \right] \right|$$

$$\leq \frac{1}{\alpha} \mathbb{E}_{x \sim p_\theta} \left[ p^2 \mathbf{1}_{C_R} \right] + R^2$$

The term $\mathbb{E}_{x \sim p_\theta} \left[ p^2(x) \mathbf{1}_{C_R} \right]$ can be bounded using Cauchy-Schwarz as follows

$$\mathbb{E}_{x \sim p_\theta} \left[ p^2(x) \mathbf{1}_{C_R} \right]^2 \leq \mathbb{E}_{x \sim p_\theta} \left[ p^4(x) \right] \cdot \mathbb{E}_{x \sim p_\theta} \left[ \mathbf{1}_{C_R} \right] \leq C \exp(-cLR),$$

where $C$ is a constant. To obtain the constant $C$, we use Proposition 6 (b), which gives

$$\mathbb{E}_{x \sim p_\theta} \left[ p^4 \right] \leq (4K_2)^4$$

Substituting back into the original inequality, we get a bound of the form

$$O \left( \frac{1}{\alpha} \exp \left( -\frac{cL}{2R} \right) \right) + R^2.$$

We used the big O notation to suppress any constants, which we may eliminate by paying only $\log(\text{constant})$. Therefore, since the above holds true for all $R$, we may minimize it or choose an $R$ that is satisfactory for our purposes. We may choose $R = O(\log^2(1/\alpha))$. Therefore,

$$O \left( \frac{1}{\alpha} \exp \left( -\frac{cL}{2} \cdot R \right) \right) + R^2 = O \left( \log^2 \left( \frac{1}{\alpha} \right) \right),$$

so we conclude.

### C.4 THE PROOF OF OBSERVATION 11

**Observation 28** (Mass along the sub-level sets). *Let $p_\theta$ be some exponential family and define $\theta'$ to be a minimizer of the constrained objective $\{\min_\theta \mathcal{L}_S(\theta) : s.t. \mathcal{L}(\theta) \leq \bar{L}\}$ for some $\bar{L}$. For any $\theta$ such that $\mathcal{L}(\theta) \leq \bar{L}$, it holds $\mathbb{P}_\theta[S] \geq \mathbb{P}_{\theta'}[S] \exp(\mathcal{L}(\theta') - \mathcal{L}(\theta))$.*

*Proof.* Since $\theta'$ is the minimizer of the constrained optimization problem $\bar{L}$, we immediately deduce that $\mathcal{L}_S(\theta') \leq \mathcal{L}_S(\theta)$. Interestingly, this result is purely a consequence of this observation. Specifically, by rearranging terms, we obtain: $\log(\mathbb{P}_{\theta'}[S]) \leq \log(\mathbb{P}_\theta[S]) + \mathcal{L}(\theta) - \mathcal{L}(\theta')$. Applying the exponential function to both sides yields: $\mathbb{P}_{\theta'}[S] \exp(\mathcal{L}(\theta') - \mathcal{L}(\theta)) \leq \mathbb{P}_\theta[S]$. Thus, the result follows. $\square$

### C.5 THE PROOF OF COROLLARY 12

**Corollary 29** (Mass monotonicity along sub-level thresholds). *Let $\theta^*$ be the minimizer of the truncated NLL objective $\mathcal{L}_S$ for some exponential family $p_\theta$, constrained on the convex domain $\{\theta : \mathcal{L}(\theta) \leq \mathcal{L}(\theta^*)\}$. Denote by $\alpha = \mathbb{P}_{\theta^*}[S]$ the mass assigned to the survival set $S$. Define the following constrained optimization problem: $\{\min_\theta \mathcal{L}_S(\theta) : s.t. \mathcal{L}(\theta) \leq \bar{L}\}$ and denote by $\theta_1$ its solution with $\bar{L} = L_1$ and by $\theta_2$ its solution of the same optimization problem with $\bar{L} = L_2$, where $L_2 > L_1$.*

*1. (Monotonicity) The mass assigned to $S$ is decreasing as a function of $\bar{L}$: $\mathbb{P}_{\theta_1}[S] \geq \mathbb{P}_{\theta_2}[S] \geq \alpha$.*

*2. (Exponential decrease) The mass assigned to $S$ by $\theta_2$ drops exponentially fast, i.e., it holds that $\mathbb{P}_{\theta_2}[S] \leq \mathbb{P}_{\theta_1}[S] \, e^{-(L_2-L_1)}$.*

*Proof.* Notice that by definition of $\theta_1$, $\theta_1 \in \{\min_\theta \mathcal{L}_S(\theta) : s.t. \mathcal{L}(\theta) \leq L_2\}$. Hence, $\mathbb{P}_{\theta_1}[S] \geq \mathbb{P}_{\theta_2}[S] \exp(L_2 - L_1) \geq \mathbb{P}_{\theta_2}[S]$. It only remains to show that $\mathbb{P}_{\theta_2}[S] \geq \alpha$. Since, the $\min_\theta \{\mathcal{L}_S(\theta) : s.t. \mathcal{L}(\theta) \leq L_2\}$ decreases as $L_2$ increases, $\mathcal{L}(\theta^*) = \min_\theta \{\mathcal{L}_S(\theta) : s.t. \mathcal{L}(\theta) < \infty\} = \min_\theta \{\mathcal{L}_S(\theta) : s.t. \mathcal{L}(\theta) \leq \mathcal{L}(\theta^*)\}$. Which means for $\theta'$ that is the solution to a constrained optimization problem with lower $\bar{L} = L_2$, we have $\theta' \in \{\mathcal{L}_S(\theta) : s.t. \mathcal{L}(\theta) \leq \mathcal{L}(\theta^*)\}$. Hence, $\bar{L} \leq \mathcal{L}(\theta^*)$ and from the previous part this implies $\alpha = \mathbb{P}_{\theta^*}[S] \leq \mathbb{P}_{\theta'}[S]$. Hence, we conclude. $\square$

### C.6 THE PROOF OF LEMMA 14

**Lemma 30** (Bounded gaps and proximity of a $(\delta, \epsilon)$ minimum). *Let $\theta_L$ the solution $\theta_L = \arg\min_{\theta \in \Theta} \mathcal{L}_S(\theta)$ such that $\mathcal{L}(\theta) \leq L$. Suppose $\overline{\theta}_L$ approximates the loss by $\epsilon$, i.e. $\mathcal{L}_S(\overline{\theta}_L) - \mathcal{L}_S(\theta_L) \leq \epsilon$. Denote $\delta = L - L(\overline{\theta}_L)$, and set $\theta_{L-\delta}$ the solution $\theta_L = \arg\min_{\theta \in \Theta} \mathcal{L}_S(\theta)$ such that $\mathcal{L}(\theta) \leq L - \delta$. Then, the following approximation is true $e^{-\epsilon-\delta}\mathbb{P}_{x \sim p_{\overline{\theta}_L}}(S) \leq \mathbb{P}_{x \sim p_{\theta_L}}[x \in S] \leq e^{-\delta}\mathbb{P}_{x \sim p_{\overline{\theta}_L}}[x \in S]$. Also, $\mathcal{L}_S(\theta_{L-\delta}) - \mathcal{L}_S(\theta_L) \leq \epsilon$. Lastly, for $g = \mathcal{O}((\frac{L}{\lambda}\log^2(1/\alpha))^{-1})$, we get $\mathcal{L}_S(\theta_L) - \mathcal{L}_S(\theta_{L+g}) \leq 1$.*

*Proof.* We get the upper bound $\mathbb{P}_{x \sim p_{\theta_L}}[x \in S] \leq e^{-\delta}\mathbb{P}_{x \sim p_{\overline{\theta}_L}}[x \in S]$ from $\mathcal{L}_S(\theta_L) \leq \mathcal{L}_S(\overline{\theta}_L)$, after rearranging. And the lower bound $e^{-\epsilon-\delta}\mathbb{P}_{x \sim p_{\overline{\theta}_L}}[x \in S] \leq \mathbb{P}_{x \sim p_{\theta_L}}[x \in S]$ from $\mathcal{L}_S(\overline{\theta}_L) \leq \mathcal{L}_S(\theta_L) + \epsilon$, after rearranging.

For the inequality $\mathcal{L}_S(\theta_{L-\delta}) - \mathcal{L}_S(\theta_L) \leq \epsilon$, just notice that $\mathcal{L}_S(\theta_{L-\delta}) \leq \mathcal{L}_S(\overline{\theta}_L)$.

Lastly, to show $\mathcal{L}_S(\theta_L) - \mathcal{L}_S(\theta_{L+g}) \leq 1$ we will use that the function $\mathcal{L}_S(\theta)$ is $L_S := L(1 + \log(1/\alpha))$-smooth.

Denote, for simplicity, $\theta_0 = \theta_L$ and $\theta_1 = \theta_{L+g}$ By smoothness we obtain,

$$\mathcal{L}_S(\theta_0) - \mathcal{L}_S(\theta_1) \leq \nabla\mathcal{L}_S^T(\theta_1)(\theta_0 - \theta_1) + \frac{L_S}{2}\|\theta_1 - \theta_0\|_2^2.$$

$$\leq \left(\nabla\mathcal{L}_S^T(\theta_1) - \nabla\mathcal{L}_S^T(\theta_0)\right)(\theta_0 - \theta_1) + \nabla\mathcal{L}_S^T(\theta_0)(\theta_0 - \theta_1) + \frac{L_S}{2}\|\theta_1 - \theta_0\|_2^2. \tag{5}$$

We estimate now the term $\nabla\mathcal{L}_S^T(\theta_0)$, since $(\theta_1 - \theta_0)^T\nabla\mathcal{L}(\theta_0) \leq 0$, it suffices to estimate the term $\frac{\nabla_\theta \mathbb{P}_{x \sim p_{\theta_0}}[x \in S]}{\mathbb{P}_{x \sim p_{\theta_0}}[x \in S]}$. We have the following,

$$\frac{\nabla_\theta \mathbb{P}_{x \sim p_{\theta_0}}[x \in S]}{\mathbb{P}_{x \sim p_{\theta_0}}[x \in S]} = \frac{\int_S \left[h(x)\exp(\theta_0^T T(x) - A(\theta_0))T(x) - h(x)\exp(\theta_0^T T(x) - A(\theta_0))A'(\theta_0)\right]\mathbf{x}}{\mathbb{P}_{x \sim p_{\theta_0}}[x \in S]}$$

$$= \frac{\int_S \left[h(x)\exp(\theta_0^T T(x) - A(\theta_0))\left(T(x) - A'(\theta_0)\right)\right]\mathbf{x}}{\mathbb{P}_{x \sim p_{\theta_0}}[x \in S]}$$

$$= \mathbb{E}_{x \sim p_{\theta_0}^S}\left[T(x) - A'(\theta_0)\right]$$

$$= \mathbb{E}_{x \sim p_{\theta_0}^S}\left[T(x)\right] - \mathbb{E}_{x \sim p_{\theta_0}}\left[T(x)\right]$$

so by Lemma 9 we obtain

$$\|\mathcal{L}_S(\theta_0)\|_2 \leq O\left(\log\left(\frac{1}{\mathbb{P}_{x \sim p_{\theta_0}}[x \in S]}\right)\right) \leq O\left(\log\left(\frac{1}{\alpha}\right)\right).$$

Going back to Equation (5), by the previous deduction and smoothness, we find

$$\mathcal{L}_S(\theta_0) - \mathcal{L}_S(\theta_1) \leq L_S\|\theta_0 - \theta_1\|_2^2 + \log(1/\alpha)\|\theta_1 - \theta_0\| + \frac{L_S}{2}\|\theta_1 - \theta_0\|_2^2,$$

therefore, by strong convexity of $\mathcal{L}(\theta)$

$$\mathcal{L}_S(\theta_0) - \mathcal{L}_S(\theta_1) \leq \frac{2L_S}{\lambda}(\mathcal{L}(\theta_1) - \mathcal{L}(\theta_0)) + \log(1/\alpha)(\mathcal{L}(\theta_1) - \mathcal{L}(\theta_0))^{1/2} + \frac{L_S}{\lambda}(\mathcal{L}(\theta_1) - \mathcal{L}(\theta_0))$$

$$\leq \log(1/\alpha)(\mathcal{L}(\theta_1) - \mathcal{L}(\theta_0))^{1/2} + O\left(\frac{L_S}{\lambda}(\mathcal{L}(\theta_1) - \mathcal{L}(\theta_0))\right)$$

Hence we conclude.

$\square$

## C.7 THE PROOF OF THEOREM 26

*Proof.* We aim to show that the sequence $\{w^{(t)}\}$ generated by the algorithm converges to an optimal point $w^*$.

We use the non-expansiveness of the projection operator. Namely, the projection operator onto a convex set is non-expansive, meaning for any $x$ and $y$,

$$\|\Pi_{\mathcal{K}}(x) - \Pi_{\mathcal{K}}(y)\| \leq \|x - y\|.$$

Using the convexity and smoothness properties, and the non-expansiveness of the projection operator, we can see the progress achieved after each iteration.

$$\|w^{(t)} - w^*\|^2 \leq \|w^{(t-1)} - h_t g^{(t)} - w^*\|^2.$$

Expanding the right-hand side, we get:

$$\|w^{(t)} - w^*\|^2 \leq \|w^{(t-1)} - w^*\|^2 - 2h_t\langle g^{(t)}, w^{(t-1)} - w^*\rangle + h_t^2\|g^{(t)}\|^2.$$

Since $f$ is convex, we have:

$$f(w^{(t-1)}) - f(w^*) \leq \langle \nabla f(w^{(t-1)}), w^{(t-1)} - w^*\rangle.$$

Taking expectations conditioned on $w^{(t-1)}$, and noting that $\mathbb{E}\left[g^{(t)} \mid w^{(t-1)}\right] = \nabla f(w^{(t-1)})$, we get:

$$\mathbb{E}\left[f(w^{(t-1)})\right] - f(w^*) \leq \mathbb{E}\left[\langle \nabla f(w^{(t-1)}), w^{(t-1)} - w^*\rangle\right].$$

Combining everything together, taking expectations, and summing over $t = 1$ to $T$,

$$\sum_{t=1}^{T}\mathbb{E}\left[f(w^{(t-1)})\right] - Tf(w^*) \leq \sum_{t=1}^{T}\mathbb{E}\left[\langle \nabla f(w^{(t-1)}), w^{(t-1)} - w^*\rangle\right]$$

$$\leq \sum_{t=1}^{T}\mathbb{E}\left[\langle g^{(t)}, w^{(t-1)} - w^*\rangle\right]$$

$$\leq \frac{1}{2L}\sum_{t=1}^{T}\mathbb{E}\left[\|w^{(t-1)} - w^*\|^2 - \|w^{(t)} - w^*\|^2\right] + h_t^2\mathbb{E}\left(\|g^{(t)}\|^2\right).$$

Using the boundedness of $\mathbb{E}\left[\|g^{(t)}\|^2\right] \leq b$, and that $h_t = \frac{1}{Lt}$, we have:

$$\sum_{t=1}^{T}h_t^2\|g^{(t)}\|^2 \leq b\sum_{t=1}^{T}\frac{1}{L^2t^2} \leq \frac{b}{L^2}\sum_{t=1}^{T}\frac{1}{t^2} \leq \frac{b}{L^2}\left(\frac{\pi^2}{6}\right).$$

So,

$$\frac{1}{T}\sum_{t=1}^{T}\mathbb{E}\left[f(w^{(t-1)})\right] - f(w^*) \leq \frac{\|w^{(0)} - w^*\|^2}{2LT} + \frac{b\pi^2}{12L^2T}.$$

Finally, from convexity we obtain

$$\mathbb{E}\left[\frac{1}{T}\sum_{t=1}^{T}f(w^{(t-1)})\right] \leq \frac{1}{T}\sum_{t=1}^{T}\mathbb{E}\left[f(w^{(t-1)})\right],$$

hence we conclude. $\qquad\square$

## C.8 THE PROOF OF LEMMA 27

Let $v$ denote the output of the rejection sampling procedure to find an unbiased estimate of the gradient. We have

$$\mathbb{E}[\|v\|^2]$$

$$= \mathbb{E}_{z \sim p_\theta^S} \mathbb{E}_{x \sim p_{\theta*}^S} \left[ \| T(z) - [T(x)] \|^2 \right]$$

$$= \mathbb{E}_{z \sim p_\theta^S} \mathbb{E}_{x \sim p_{\theta*}^S} \left[ \| T(z) \|^2 - 2T(z)^\top T(x) + \| [T(x)] \|^2 \right]$$

$$= \text{Tr}(\text{Cov}[T(z)]) + (\mathbb{E}[\|T(z)\|])^2 + \text{Tr}(\text{Cov}[T(x)]) + (\mathbb{E}[\|T(x)\|])^2 - 2\langle \mathbb{E}[T(z)], \mathbb{E}[T(x)]$$

$$= \text{Tr}(\text{Cov}[T(z)]) + \text{Tr}(\text{Cov}[T(x)]) + \| \mathbb{E}_{x \sim p_{\theta_i}^S}[T(z)] - \mathbb{E}_{x \sim p_{\theta*}^S}[T(x)] \|^2$$

$$\le k(L + \log \left( \tfrac{1}{\alpha} \right)) + kL + O \left( \left( \lambda^{-1} (L + \log^2 \left( \tfrac{1}{\alpha} \right)) \right)^2 \log^2 \left( \tfrac{1}{\alpha} \right) \right).$$

Since,

$$\| \mathbb{E}_{x \sim p_{\theta_i}^S}[T(z)] - \mathbb{E}_{x \sim p_{\theta*}^S}[T(x)] \| = \| \nabla \mathcal{L}_S(\theta_i) - \nabla \mathcal{L}_S(\theta^*) \|$$

$$\le O \left( \left( L + \log^2 \left( \frac{1}{\alpha} \right) \right) \right) \| \theta_i - \theta^* \|$$

$$\le O \left( \lambda^{-1} \left( L + \log^2 \left( \frac{1}{\alpha} \right) \right) \right) \log \left( \frac{1}{\alpha} \right)$$

where we used Lemma 9 in combination with the assumption of strong convexity of $\mathcal{L}(\theta)$ and the upper bound of the smoothness of $\mathcal{L}_S(\theta)$.

# D    PROOFS FROM SECTION 5

## D.1    PROOF OF LEMMA 19

**Lemma 31** (Closed Under Mixtures Implies Powerful). *If a family $F$ is $\alpha$-closed under mixtures with respect to $S$, then $F$ is $\alpha$-powerful with respect to $S$.*

*Proof.* Assume that $F$ is $\alpha$-closed under mixtures with respect to $S$. Let $p \in F$ with $p(S) > \alpha$, and let $p_m$ be the mixture distribution defined in definition 18, such that $p_m(S) = \alpha$.

For a single observation $x \in X$, consider the ratio:

$$\frac{p(x)}{p_m(x)} = \begin{cases} 1, & \text{if } x \in X \setminus S, \\ \dfrac{p(S)}{\alpha}, & \text{if } x \in S. \end{cases}$$

Taking the natural logarithm:

$$\ln \left( \frac{p(x)}{p_m(x)} \right) = \begin{cases} 0, & \text{if } x \in X \setminus S, \\ \ln \left( \frac{p(S)}{\alpha} \right), & \text{if } x \in S. \end{cases}$$

Compute the right-hand side (RHS) of the inequality in definition 17 for a single observation:

$$\exp \left( \ln \left( \frac{p(x)}{p_m(x)} \right) \right) = \frac{p(x)}{p_m(x)}.$$

Compute the left-hand side (LHS): $\frac{p_m(S)}{p(S)} = \frac{\alpha}{p(S)}$.

We need to show that: $\frac{p_m(S)}{p(S)} \le \frac{p(x)}{p_m(x)}$. We consider two cases, $x \in X \setminus S$ and $x \in S$. In the first case $\frac{p(x)}{p_m(x)} = 1$, $\frac{p_m(S)}{p(S)} = \frac{\alpha}{p(S)} \le 1$. Since $\frac{\alpha}{p(S)} \le 1$, the inequality holds:

$$\frac{\alpha}{p(S)} \le 1 = \frac{p(x)}{p_m(x)}.$$

In the second case

$$\frac{p(x)}{p_m(x)} = \frac{p(S)}{\alpha}, \qquad \frac{p_m(S)}{p(S)} = \frac{\alpha}{p(S)}.$$

Therefore, the inequality becomes:

$$\frac{\alpha}{p(S)} \le \frac{p(S)}{\alpha}.$$

Multiplying both sides by $\frac{p(S)}{\alpha}$: $\left(\frac{\alpha}{p(S)}\right)^2 \le 1$. Since $0 < \alpha \le p(S) \le 1$, we have $\left(\frac{\alpha}{p(S)}\right)^2 \le 1$, so the inequality holds.

To extend to the general, we use that this the wanted property holds for each observation $x_i$ by multiplying all $\frac{p_m(S)}{p(S)} \le \frac{p(x_i)}{p_m(x_i)}$ and then normalizing by taking the $n$-th root we conclude that

$$\frac{p_m(S)}{p(S)} \le \exp\left(\frac{1}{n} \sum_{i=1}^{n} (\ln p(x_i) - \ln p_m(x_i))\right).$$

$\square$

## D.2   PROOF OF COROLLARY 20

**Corollary 32** (Exponential Family is $\alpha$-Closed Under Mixtures). *Let $\{p_\theta\}_{\theta \in \Theta}$ be an exponential family of probability density functions on a space $X$, with sufficient statistic $T(x)$ and base measure $h(x)$. Then, for any $\alpha \in (0, 1]$, the exponential family is $\alpha$-closed under mixtures with respect to any measurable subset $S \subset X$.*

*Proof.* We need to show that for any $p_\theta \in F$ with $p_\theta(S) > \alpha$, the mixture distribution $p_m$ defined in Definition 18 is also in the exponential family.

Consider the extended space $X' = X \cup \{x^*\}$. Define the extended sufficient statistic $\tilde{T}(x)$ and base measure $\tilde{h}(x)$ as:

$$\tilde{T}(x) = \begin{cases} T(x), & \text{if } x \in X, \\ 0, & \text{if } x = x^*, \end{cases} \qquad \tilde{h}(x) = \begin{cases} h(x), & \text{if } x \in X, \\ h(x^*), & \text{if } x = x^*, \end{cases}$$

where $h(x^*) > 0$ is finite.

Define the extended natural parameter $\tilde{\theta} = (\theta, t)$, where $t \in \mathbb{R}$ is an additional scalar parameter controlling the mass at $x^*$.

The extended density function $q_{\tilde{\theta}}(x)$ is given by:

$$q_{\tilde{\theta}}(x) = \tilde{h}(x) \exp\left(\theta^\top \tilde{T}(x) + t \cdot 1_{\{x=x^*\}} - A(\tilde{\theta})\right),$$

where $1_{\{x=x^*\}}$ is the indicator function, and $A(\tilde{\theta})$ is the log-partition function ensuring normalization:

$$A(\tilde{\theta}) = \ln\left(\int_X h(x) \exp\left(\theta^\top T(x)\right) dx + h(x^*) \exp(t)\right).$$

By adjusting $t$, we can set $q_{\tilde{\theta}}(S) = \alpha$. Specifically, we solve for $t$ such that:

$$q_{\tilde{\theta}}(S) = \int_S q_{\tilde{\theta}}(x) dx = \alpha.$$

Since $q_{\tilde{\theta}}$ remains within the exponential family for all $\theta \in \Theta$ and $t \in \mathbb{R}$, and the mixture distribution $p_m$ corresponds to a particular choice of $\tilde{\theta}$, it follows that $p_m \in F$. Therefore, the exponential family is $\alpha$-closed under mixtures with respect to $S$. $\square$

**Remark:** By selecting $t$ to be a finite but negative number enough, we can ensure that the probability assigned to $x^*$ is extremely small. Specifically, we choose $t$ such that: $h(x^*) \exp(t) = \varepsilon'$, with $\epsilon' \ll 1$. The log-partition function becomes: $A(\tilde{\theta}) = \ln(Z(\theta) + \varepsilon') \approx A(\theta)$, since $\varepsilon'$ is negligible compared to $Z(\theta)$.

The total variation distance between $p_\theta$ and $q_{\tilde\theta}$ is then:

$$\text{TV}(p_\theta, q_{\tilde\theta}) = \frac{1}{2}\left(\int_X \left|p_\theta(x) - q_{\tilde\theta}(x)\right| dx + \left|0 - q_{\tilde\theta}(x^*)\right|\right)$$

$$\leq \frac{1}{2}\left(\varepsilon'' + \varepsilon'\right) < \varepsilon, \text{ for some small } \epsilon > 0$$

Therefore, $q_{\tilde\theta}$ is $\epsilon$-close to $p_\theta$ in total variation distance, and the inequality in Definition 17 holds with negligible difference. In other words we may assume that our distributions exist in the same parameter space.

### D.3 PROOF OF PROPOSITION 24

**Proposition 33** (Minima lie near the boundary). *Suppose that $\{p_\theta\}_{\theta\in\Theta}$ forms an exponential family that is powerful (see Definition 17) with respect to a set $S \subset X$, taking values in $X$. Let $v$ be a vector satisfying $u^T v \geq a$, where $u$ is the direction of decrease as defined in proposition 22. Define $A \subset \Theta$ as the domain*

$$A = \{\theta \in \Theta : \mathcal{L}(\theta) - \mathcal{L}(\theta_0) \leq M\},$$

*and set $\theta_M^* = \arg\min_{\theta\in A}\mathcal{L}(\theta)$. Then, for a $\tilde\epsilon$-minimizer $\theta_M$ of $\mathcal{L}_S^{\epsilon',v}$ over $A$, where $\tilde\epsilon = O(\epsilon'\epsilon a \log(1/p_{\theta_M^*}(S)))$ such that $\theta_M$ satisfies*

$$M + \mathcal{L}(\theta_0) - \epsilon \leq \mathcal{L}(\theta_M),$$

*where $\mathcal{L}_S^{\epsilon',v}$ is the $\epsilon'$-regularized loss function (see definition 23), and $\mathcal{L}(\theta)$ is the negative log-likelihood of the exponential family.*

*Proof.* Fix $\epsilon > 0$. By the assumption that the family $\mathcal{F}$ is powerful, we invoke Proposition 22 and take a unit vector $v$ such that $v^T u \geq \delta/d \triangleq a$, where $v$ is such that for all $\theta' \in \{\theta \in \Theta : \mathcal{L}(\theta) - \mathcal{L}(\theta_0) \leq \log(1/\alpha)\}$, the gradient of $\mathcal{L}_S$ satisfies $u^T \nabla\mathcal{L}_S(\theta') \leq 0$.

Suppose that we pick $\theta_M$ such that

$$\mathcal{L}_S(\theta_M) < \mathcal{L}_S(\theta_M^*) + \tilde\epsilon,$$

where $\theta_M^*$ is the minimizer of $\mathcal{L}_S$ on the domain $A_M = \{\theta : \mathcal{L}(\theta) - \mathcal{L}(\theta_0) \leq M\}$. From lemma 9, we know that $\mathcal{L}_S$ is $L_S$-smooth with

$$\mathcal{L}_S = O\left(L\left(1 + \log^2\left(1/\mathbb{P}_{x\sim p_{\theta_M^*}}[x \in S]\right)\right)\right).$$

A standard property for this class of functions is (see lemma 2.25 in (Garrigos & Gower, 2024)):

$$f(y) \leq f(x) + \nabla f(x)^T(y - x) + \frac{L_S}{2}\|y - x\|^2.$$

Rearranging and setting $f, y, x = \mathcal{L}_S, \theta, \theta_M$, respectively, we obtain the following:

$$(\theta - \theta_M)^T\nabla\mathcal{L}_S^{\epsilon,v}(\theta_M) + \frac{L_S^{\epsilon,v}}{2}\|\theta - \theta_M\|^2 \geq \mathcal{L}_S^{\epsilon,v}(\theta) - \mathcal{L}_S^{\epsilon,v}(\theta_M) \geq \mathcal{L}_S^{\epsilon,v}(\theta_M^*) - \mathcal{L}_S^{\epsilon,v}(\theta_M) \geq -\tilde\epsilon. \quad (6)$$

The above holds for all $\theta \in A_M$. On the other hand, for $\theta - \theta_M = \gamma u$, where $\gamma = \max\{\gamma : \gamma \in [0, \infty) : \gamma u + \theta_M \in A_M\}$ and $u$ is the unit direction of decrease as described in Proposition 22. Therefore, we find

$$(\theta - \theta_M)^T\nabla\mathcal{L}_S^{\epsilon,v}(\theta_M) = (\gamma u)^T(\nabla\mathcal{L}_S(\theta_M) - \epsilon'v)$$
$$= (\gamma u)^T(\nabla\mathcal{L}_S(\theta_M) - \epsilon'v)$$
$$= \gamma u^T\nabla\mathcal{L}_S(\theta_M) - \epsilon'\gamma a$$
$$\leq -\epsilon'a\gamma.$$

Substituting back into Equation (6), we get $-\tilde\epsilon \leq -\epsilon'a\gamma + \frac{L_S^{\epsilon,v}}{2}\gamma^2$. Therefore,

$$\gamma \leq \frac{\varepsilon'a - \sqrt{(\varepsilon'a)^2 - 2L_S^\varepsilon\tilde\varepsilon}}{L_S^\varepsilon}$$

or approximately, $\gamma \leq \frac{\tilde{\epsilon}}{\epsilon' a}$. However, there exists $\theta_b$ such that $\mathcal{L}(\theta_b) = M + \mathcal{L}(\theta_0)$ and $\|\theta_b - \theta_M\|_2 \leq \gamma$. Moreover,

$$\left|\mathcal{L}(\theta_b) - \mathcal{L}(\theta_M)\right| \leq \frac{L}{2\lambda} \log(1/\alpha) \|\theta_b - \theta_M\|_2,$$

so

$$M + \mathcal{L}(\theta_0) - \frac{L}{2\lambda} \log(1/\alpha) \left(\frac{\tilde{\epsilon}}{\epsilon' a}\right) \leq \mathcal{L}(\theta_M).$$

Our choice of $\tilde{\epsilon}$ is $\epsilon' a \log\left(1/\mathbb{P}_{x \sim p_{\theta_M^*}}[x \in S]\right)^{-1} \epsilon$, so $M + \mathcal{L}(\theta_0) - \epsilon \leq \mathcal{L}(\theta_M)$. $\qquad\square$

# E   NP-HARDNESS OF SOLVING EQUATION (3) WITHOUT ADDITIONAL ASSUMPTIONS

To show NP-hardness of (3), we will make a reduction to a variant of **MAX-CUT**. A common formulation of the **MAX-CUT** problem is to find a partition that achieves the cut score prescribed in the input. To be more precise, let $G = (V, E)$ be a graph with $d$ vertices. A cut is a partition of $V$ into two subsets $S$ and $S^c = V \setminus S$ and the value of the cut $(S, S^c)$ is $c(S, S^c) = \sum_{(u,v) \in E} 1\{u \in S, v \in S^c\}$.

An equivalent formulation of the **MAX-CUT** is the following maximization problem, given the same set-up as before.

$$\text{find} \quad \sum_{(i,j) \in E} (x_i - x_j)^2 = \text{OPT}, \text{ subject to} \quad x_i \in \{-1, +1\} \quad \forall i \in [d]. \tag{7}$$

It is easy to verify the more succinct version of the objective

$$x^T L_G x = \sum_{(i,j) \in E} (x_i - x_j)^2$$

where $L_G$ is the Laplacian matrix of the graph $G$.

Furthermore, the reduction will use the NP-hardness for any close enough approximation to the **MAX-CUT** problem. Namely, we quote Hastad (2001) about the inapproximability of the MAX-Cut problem.

**Lemma 34** (Inapproximability of **MAX-CUT** Hastad (2001)). *It is NP-hard to approximate **MAX-CUT** to any factor higher than $16/17$.*

To solve (7), one constraint is that the feasible set should be on the hypercube, and secondly, that the term

$$x^T L_G x = \sum_{(i,j) \in E} (x_i - x_j)^2$$

is approximately maximized. Denote $H_d = \{\pm 1\}^d$ the hypercube in $d$-dimensions. Also, set

$$H_d^\epsilon = H_d + B(\epsilon),$$

the $\epsilon$-neighborhood of the hypercube, where $B(\epsilon) = \{x \in \mathbb{R}^d : \|x\| = \epsilon\}$.

The set $S$ from the optimization problem (3) is defined as follows

$$S = \left(H_d^\epsilon \bigcap \left\{\frac{x^T L_G x}{OPT} > \frac{33}{34}\right\}\right)^c,$$

where the constant $\epsilon$ is chosen such that, whenever, $x, y \in H_d^\epsilon$ such that $\|x - y\| < \epsilon$ then,

$$|x^T L_G x - y^T L_G y| < \frac{OPT}{34}.$$

For the previous fixed $\epsilon$ we choose $\delta$ small enough so that

$$N(\theta, \delta; S) > \alpha$$

whenever $\theta \in S$.

In order to find a solution of the optimization problem (3) it is required first to locate a $\theta$, such that $\mathbb{P}_{x \sim p_\theta}[x \in S] \leq \alpha$. On the other hand, any such $\theta$ must lie in the complement of $S$, i.e. $\theta \in S^c$.

Denote, $\tilde{\theta}$ the solution of

$$\tilde{\theta} := \arg\min_{\theta' \in H_d} \|\theta' - \theta\|.$$

Since $\left\|\tilde{\theta} - \theta\right\| < \epsilon$, we see that

$$\tilde{\theta}^T L_G \tilde{\theta} > \theta L_G \theta - \frac{OPT}{34} > \frac{16}{17} OPT.$$

Hence, we conclude the result for the family $N(\theta, \delta)$. However, by simple change of coordinates we may rescale our family to be $N(\theta, I)$.

# F    COMPLETE PROOF OF THEOREM 16

We present the full proof of Theorem 16 in two parts.

**Part I** treats the "no-overshoot" regime, where the outer search does not cross the $\alpha$ threshold and termination is triggered by the small successive-sublevel gap; this is the critical case and drives the main bounds.

**Part II** handles the complementary case where the procedure crosses $\alpha$ and invokes the binary search, showing it returns a parameter with the desired truncated loss and mass guarantees. Across both parts, we use Proposition 24 to pass to a truncated loss $\mathcal{L}_S$ that preserves boundary behavior, and we apply the PSGD guarantee plus a standard amplification step to obtain high-probability $\epsilon$-accurate solutions on each sublevel.

## F.1    FIRST PART OF THE PROOF OF THEOREM 16

Proposition 24 allows us to assume that we can work with the function $\mathcal{L}_S$ while inheriting the property that its approximate minima lie on the boundary. Indeed, we may choose $\epsilon$ such that $\sup_{\theta \in A} |\mathcal{L}_S^{\epsilon,v}(\theta) - \mathcal{L}_S(\theta)| \leq \epsilon'$ as long as $A$ is bounded, which is the case for us. The penalty we incur for choosing $\epsilon$ is linear in terms of $\tilde{\epsilon}$, as shown in Proposition 24.

Furthermore, to ensure this property holds, we will need to restart the optimization process $\Theta(1)$ times to find a vector $v$ such that $u^T v \geq a/\sqrt{k}$, where $k$ is the dimension of $\theta \in \Theta$ and $u$ is the direction of increase of $\mathcal{L}_S$ as defined in Proposition 22.

The parameter $\theta$ that will satisfy the properties of the theorem will be the output of the Algorithm 1. We will distinguish two main cases, depending on which stopping condition is triggered. However, before handling this, we first show that there is a way to amplify the probability of the output $\theta_{\bar{L}}$ as it appears on line 3, by running the PSGD independently and then choosing the best output among all the trials.

Apply Theorem 26 (PSGD) to the optimization problem Equation (4), for the sublevel set $\bar{L} = L_{\min} + mg$ for $m \in [0, \log\left(\frac{1}{\alpha}\right)/g]$. We have the following bound:

$$\|\theta_0 - \theta_\alpha^*\|_2^2 \leq \frac{2}{\lambda}\left(\mathcal{L}(\theta_0) - \mathcal{L}(\theta_\alpha^*)\right) \leq O\left(\frac{1}{\lambda}\log\left(\frac{1}{\alpha}\right)\right)$$

Hence, Theorem 26 in our setup gives an upper bound: $\frac{C \log^2(1/\alpha)}{2(L+\log(1/\alpha))T} + \frac{b\pi^2}{12(L+\log(1/\alpha))^2T}$. So, for: $T \geq \frac{1}{\epsilon}\left(\frac{C\log^2(1/\alpha)+6(L+\log(1/\alpha))b\pi^2}{12(L+\log(1/\alpha))^2}\right)$, Theorem 26 gives us a $\bar{\theta}$ such that: $\mathbb{E}\left[f(\bar{\theta})\right] - f(\theta^*) \leq \epsilon$. From Markov's inequality, we get: $\mathbb{P}\left[f(\bar{\theta}) - f(\theta^*) \geq 3\epsilon\right] \leq \frac{1}{3}$ We can easily amplify this probability by repeating this process independently and hence obtaining a sequence of $\bar{w}_1, \bar{w}_2, \ldots, \bar{w}_m$, and then choosing: $\bar{w} = \arg\min_{\bar{w}_i} f(\bar{w}_i)$. Since $\bar{\theta} := \bar{w}$ satisfies: $\mathbb{P}\left(f(\bar{\theta}) - f(\theta^*) \geq 3\epsilon\right) \leq \left(\frac{1}{3}\right)^m$, by choosing $m \geq \log(\delta)/\log(1/3)$, we obtain an $\bar{w}$ that satisfies: $\mathbb{P}\left(f(\bar{\theta}) - f(\theta^*) \geq 3\epsilon\right) \leq \delta$. To accomplish this, however, we need access to the value of: $\mathcal{L}_S(\bar{w}_i) = \mathcal{L}(\bar{w}_i) + \log\left(\mathbb{P}_{x \sim p_{\bar{w}_i}}[x \in S]\right)$.

Since we have access to $S$ only through its oracle, in order to calculate $\log\left(\mathbb{P}_{x\sim p_{\overline{w}_i}}[x\in S]\right)$, we use concentration for a Bernoulli random variable. More specifically, by Hoeffding's inequality, we need $O\left(\frac{1}{\epsilon}\log\left(\frac{1}{\alpha}\right)\right)$ samples to estimate $\log\left(\mathbb{P}_{x\sim p_{\overline{w}_i}}[x\in S]\right)$ $\epsilon$-close with probability at least $1-\delta$.

Therefore, for each sublevel set $\mathcal{L}_{\min}+mg$, when we perform PSGD, we have an output $\theta_m$ such that, with high probability, it achieves high precision: $\mathcal{L}_S(\theta_m)-\mathcal{L}_S(\theta_m^*)<\epsilon'$ where $\theta_m^*$ is the solution to the optimization problem Equation (4) for the sublevel set $\mathcal{L}_{\min}+mg$.

We return our attention to the time that Algorithm 1 is terminated. Suppose Algorithm 1 is terminated when the if statement on line 7 is verified. Then, the $\theta_{\bar{L}}$ generated at line 3 satisfies $\left|\mathcal{L}_S(\theta_{\bar{L}-g})-\mathcal{L}_S(\theta_{\bar{L}})\right|\leq\epsilon''$. Fix $\epsilon>0$ such that $\epsilon<g$, and set $\epsilon'=\epsilon''=\frac{g\epsilon}{\log(1/\alpha)}=O\left(\frac{\lambda}{L}\epsilon\frac{1}{\log^3(1/\alpha)}\right)$. We claim that for this choice of $\epsilon''$, Algorithm 1 produces an output $\theta_f$ with high probability, such that $\mathcal{L}_S(\theta_f)-\mathcal{L}_S(\theta_\alpha^*)\leq\epsilon$.

Suppose that Algorithm 1 is terminated at line 7 of Algorithm 1 at iteration $m$. First, we argue that $\theta_\alpha^*\notin\{\theta:\mathcal{L}(\theta)-L_{\min}\leq(m-1)g\}$. We argue by contradiction, namely suppose that $\theta_\alpha^*\in\{\theta:\mathcal{L}(\theta)-L_{\min}\leq(m-1)g\}$. From Observation 11 we get $\mathbb{P}_{x\sim p_{\theta_{m-1}}}[x\in S]\leq\alpha$, which in turn implies $\mathbb{P}_{\theta_m}(S)\leq\alpha\exp(-g)$. On the other hand, since Algorithm 1, at iteration $m$, did not terminate at line 4, it implies that $\mathbb{P}_{x\sim p_{\theta_m}}[x\in S]>a\exp(\delta)$. But, from Lemma 14 we know that $\mathbb{P}_{x\sim p_{\theta_m}}[x\in S]\leq\mathbb{P}_{x\sim p_{\theta_m^*}}[x\in S]\exp(\epsilon'+\delta)$. Therefore, $a\exp(\delta)<\mathbb{P}_{x\sim p_{\theta_m}}[x\in S]\leq\mathbb{P}_{x\sim p_{\theta_m^*}}[x\in S]\exp(\epsilon'+\delta)\leq\alpha\exp(\epsilon'+\delta-g)<\alpha\exp(\delta)$, a contradiction.

Then $\theta_\alpha^*$ is either in $\{\theta:\mathcal{L}(\theta)-L_{\min}\leq mg\}$ or it is not. Suppose the first case, i.e. $\theta_\alpha^*\in\{\theta:\mathcal{L}(\theta)-L_{\min}\leq mg\}$. By Observation 11, $\mathcal{L}_S(\theta_{m-1})\geq\mathcal{L}_S(\theta_\alpha^*)\geq\mathcal{L}_S(\theta_m^*)$. Also, $\mathcal{L}_S(\theta_f)-\mathcal{L}_S(\theta_m^*)\leq\epsilon'$. Therefore, using $|\mathcal{L}_S(\theta_f)-\mathcal{L}_S(\theta_{m-1})|\leq\epsilon''$, we get

$$|\mathcal{L}_S(\theta_f)-\mathcal{L}_S(\theta_\alpha^*)|\leq 2\epsilon'+\epsilon'',$$

and we are done.

We deal now with the other case. From Proposition 15 $\mathcal{L}_S(\theta_{m-1}^*)-\mathcal{L}_S(\theta_m^*)\geq\frac{\mathcal{L}_S(\theta_{m-1}^*)-\mathcal{L}_S(\theta_\alpha^*)}{\log\left(\frac{1}{\alpha}\right)/g}$, and suppose we get $\theta_f=\theta_m$. From our assumption, recall that $|\mathcal{L}_S(\theta_m)-\mathcal{L}_S(\theta_{m-1})|\leq\epsilon''$, where $\epsilon''$ is our threshold value for stopping Algorithm 1. Then,

$$\mathcal{L}_S(\theta_{m-1}^*)-\mathcal{L}_S(\theta_m^*)\leq|\mathcal{L}_S(\theta_{m-1})-\mathcal{L}_S(\theta_m)|$$
$$+2\epsilon'\leq\epsilon''+2\epsilon'.$$

Therefore, by our choice of $\epsilon',\epsilon''$, we find that $\frac{\mathcal{L}_S(\theta_{m-1}^*)-\mathcal{L}_S(\theta_\alpha^*)}{\log\left(\frac{1}{\alpha}\right)/g}\leq\frac{3\epsilon g}{\log(1/\alpha)}\Rightarrow\mathcal{L}_S(\theta_f)-\mathcal{L}_S(\theta_\alpha^*)\leq\mathcal{L}_S(\theta_{m-1}^*)-\mathcal{L}_S(\theta_\alpha^*)\leq 3\epsilon$.

In the previous case, when termination occurred on line 7, we obtained a suitable output by virtue of identifying a threshold value $\epsilon''$ so small that if the difference in the values of the approximated minima between two successive sets was bounded by $\epsilon''$, then even if we had continued our search, our progress in terms of reducing the value of $\mathcal{L}_S$ would have been negligible. Finally, by Proposition 24 since the both $\mathcal{L}_S(\theta_f)$ and $\mathcal{L}(\theta_f)$ are close to the boundary we conclude that $\mathbb{P}_{x\sim\theta_f}[x\in S]$ is close to $\alpha$. What remains is the case where Algorithm 1 terminates on line 4, however that is a simple binary search and we defer the reader to Section F.2.

## F.2 SECOND PART OF THE PROOF OF THEOREM 16

We deal now with the case when Algorithm 1 is terminated when calling Algorithm 4. Suppose, that Algorithm 1 enters line 9 at iteration $m$, and denote $L_m=\bar{L}=L_{\min}+mg$, and $\theta_m^*$ the corresponding solution of Equation (4) for $\bar{L}=L_{\min}+mg$.

Therefore, by Lemma 14, we get

$$\mathbb{P}_{x\sim p_{\theta_m^*}}[x\in S]\leq\mathbb{P}_{x\sim p_{\theta_m}}[x\in S]\exp(L_m-\mathcal{L}(\theta_m))\leq\alpha,$$

hence, by Observation 11, $L_m>\mathcal{L}(\theta_\alpha^*)$. Next, we show that $\mathcal{L}(\theta_\alpha^*)>L_{m-2}$. As before, we argue by contradiction and assume $\mathcal{L}(\theta_\alpha^*)\leq L_{m-2}$, which in turn implies $\mathbb{P}_{x\sim p_{\theta_{m-2}}}[x\in S]\leq\alpha$. Indeed, as we saw before, since Algorithm 1 did not terminate on line 5 in iteration $m-1$, we get

$\mathbb{P}_{x \sim p_{\theta_{m-1}}}[x \in S] > \alpha \exp(\delta)$. Also, by Lemma 14 and Observation 11, as we argued in the previous case, we get

$$\alpha \exp(\delta - \delta - \epsilon) \leq \exp(-\delta - \epsilon)\mathbb{P}_{x \sim p_{\theta_{m-1}}}[x \in S] \leq \mathbb{P}_{x \sim p_{\theta_{m-1}}}[x \in S] \leq \alpha \exp(-g),$$

a contradiction. Hence, $\theta_\alpha \in \{\theta : L_{m-2} < \mathcal{L}(\theta) < L_m\}$. Now, in order to obtain $\bar{\theta}_l$ such that $\mathcal{L}_S(\bar{\theta}_l) - \mathcal{L}_S(\theta_\alpha)$, we call `binary search`$(L_m)$. We argue that the output of Algorithm 4 (`binary search`$(\cdot)$) indeed possesses the desired properties.

We distinguish two cases. Suppose that $\mathbb{P}_{x \sim p_{\bar{\theta}_l}}[x \in S] < \alpha$. Then, immediately, we have $\mathcal{L}(\bar{\theta}_l) \geq \mathcal{L}(\theta_\alpha^*)$. Henceforth, $\mathcal{L}_S(\bar{\theta}_l) - \mathcal{L}_S(\theta_\alpha^*) \leq \epsilon$. Furthermore, the current Left at the time of termination had an associated $\bar{\theta}_{\text{Left}}$ that satisfied $\mathbb{P}_{x \sim p_{\bar{\theta}_{\text{Left}}}}[x \in S] \geq \alpha$, hence $\mathcal{L}_S(\bar{\theta}_{\text{Left}}) \geq \mathcal{L}_S(\theta_\alpha^*)$. On the other hand, $\mathcal{L}(\bar{\theta}_l) - \mathcal{L}(\bar{\theta}_{\text{Left}}) \leq \epsilon g$, which by Lemma 14 implies that the corresponding minima differ by at most $\epsilon$. So, by the triangle inequality,

$$\mathcal{L}_S(\bar{\theta}_l) \geq \mathcal{L}_S(\bar{\theta}_{\text{Left}}) - \epsilon - \epsilon - \epsilon = \mathcal{L}_S(\bar{\theta}_{\text{Left}}) - 3\epsilon,$$

where the first $\epsilon$ comes from the previous observation, and the next two $\epsilon$ come from the fact that both $\mathcal{L}_S(\bar{\theta}_l)$ and $\mathcal{L}_S(\bar{\theta}_{\text{Left}})$ are $\epsilon$-approximate minima. Putting everything together,

$$-3\epsilon \leq \mathcal{L}_S(\bar{\theta}_l) - \mathcal{L}_S(\theta_\alpha^*) \leq \epsilon.$$

In the other case, suppose that $\mathbb{P}_{x \sim p_{\bar{\theta}_l}}[x \in S] \geq \alpha$. Therefore, as before, $\mathcal{L}_S(\bar{\theta}_l) \geq \mathcal{L}_S(\theta_\alpha^*)$. Also, the current Right at the time of termination had a corresponding $\bar{\theta}_{\text{Right}}$ that satisfied $\mathbb{P}_{x \sim p_{\bar{\theta}_{\text{Right}}}}[x \in S] \geq \alpha$, hence $\mathcal{L}_S(\bar{\theta}_{\text{Right}}) \geq \mathcal{L}_S(\theta_\alpha^*)$. Finally, similarly as before, we have $\mathcal{L}(\theta_{\text{Right}}) - \mathcal{L}(\theta_l) \leq \epsilon g$, which by Lemma 14 implies that the corresponding minima differ by at most $\epsilon$. So, by the triangle inequality, we obtain

$$0 \leq \mathcal{L}_S(\bar{\theta}_l) - \mathcal{L}_S(\theta_\alpha^*) \leq 3\epsilon,$$

So in all cases we conclude that

$$-3\epsilon \leq \mathcal{L}_S(\bar{\theta}_l) - \mathcal{L}_S(\theta_\alpha^*) \leq 3\epsilon.$$

At each $\theta \in \Theta$ in our parameter space in order to calculate, with high probability, a gradient through rejection sampling we need to make $O\left(\mathbb{P}_{x \sim p_\theta}[x \in S]\right)$ calls to the oracle. So, to upper bound the number of calls needed at each iteration, we need to identify how small $\mathbb{P}_{x \sim p_\theta}[x \in S]$ can become.

By Observation 11, it suffices to lower bound the mass of $\mathbb{P}_{x \sim p_\theta}[x \in S]$ of $\theta = \arg\min \mathcal{L}(\theta)$, $\mathcal{L}(\theta) \leq L_{\max}$, where $L_{\max}$ is the largest sublevel set that Algorithm 1 needs to access, which is the last $\bar{L}$ before the algorithm terminates, that the loop on line 1 iterates on.

Again, we distinguish two cases, depending on where Algorithm 1 terminates. First, we deal with the more direct case, i.e., where the algorithm terminates on line 7. As we have already seen, $\theta_\alpha^* \in \{\theta : \mathcal{L}(\theta) - L_{\min} > L_{m-1}\}$. If $\theta_\alpha^* \notin \{\theta : \mathcal{L}(\theta) - L_{\min} \leq L_m\}$, then all the $\theta \in \{\theta : \mathcal{L}(\theta) - L_{\min} \leq L_m\}$ satisfy $\mathbb{P}_{x \sim p_\theta}[x \in S] \geq \alpha$ or $\mathbb{P}_{x \sim p_\theta}[x \in S]^{-1} = O(\frac{1}{\alpha})$. We deal now with the case $\theta_\alpha^* \in \{\theta : \mathcal{L}(\theta) - L_{\min} \leq L_m\}$. In this case, $\mathcal{L}(\theta_\alpha^*) - \mathcal{L}(\theta_m^*) \leq 3\epsilon$ since the stopping condition implies $\mathcal{L}(\theta_{m-1}^*) - \mathcal{L}(\theta_m^*) \leq 2\epsilon + \epsilon'' \leq 3\epsilon$. Therefore, after rearranging, we obtain

$$\alpha(1 - 3\epsilon - g) < \alpha \exp(-3\epsilon - g) < \mathbb{P}_{x \sim p_{\theta_m^*}}[x \in S]$$

hence again we achieve a rate of $\mathbb{P}_{x \sim p_\theta}[x \in S] = O(1/\alpha)$ for all $\theta$ accessed in Algorithm 1.

