# OpenReview forum: "Reducing Hallucinations in Generative Models through Truncated Statistics"
_ICLR.cc/2026/Conference — Submitted to ICLR 2026_

### Official Review · Reviewer_T5Vg · 2025-10-29

**Soundness:** 3
**Presentation:** 2
**Contribution:** 3
**Rating:** 6
**Confidence:** 3

**Summary:**

This paper formulates the problem of hallucination in generative models as a constrained loss minimization task. The model is treated as a probability distribution over the data space, and the authors assume access to an invalidity oracle that determines whether a generated instance constitutes a hallucination. The learning objective is to minimize the negative log-likelihood on the training data while constraining the probability mass assigned to the invalid region to remain below a prescribed threshold.

The setting is precisely the same as the valid generative modeling framework introduced by Hanneke et al. (2018). The main technical contribution is a computationally efficient algorithm for solving this constrained optimization problem. Crucially, the result holds for a class of so-called “powerful” distribution families, which can be satisfied by any general exponential family. The proof follows by reducing the constrained optimization problem to a truncated negative log-likelihood optimization problem, which allows for efficient resolution.

**Strengths:**

1. Although I did not verify every details of the proof due to time constraints, the technical result that establishing an efficient solution to the constrained negative-log-likelihood problem over general exponential families appears both interesting and significant.

2. The paper also introduces novel conceptual and technical ideas, notably the notion of the “powerfulness” of a distribution family, which provides a useful structural condition linking expressivity to computational tractability.

**Weaknesses:**

1. In line 114, the authors claim that “the assumption is mild and purely about expressivity: **any** parametric family ….” I understand that the argument is to expand the support by adding an additional element $x^\*$ and extending the family via the construction of $p_m$. However, to establish *powerfulness*, shouldn’t the proof operate within this expanded family as well? This reasoning works for exponential families, but I do not see how it holds for arbitrary parametric families.

2. It seems to me that the proposed algorithmic approach is specifically tailored to exponential families. Can the method can be extended to more general model classes?

3. Given that the paper’s main contribution is computational efficiency, it would be helpful to include some empirical validation, which shouldn't be too hard.

4. Is the sentence in line 200 generated by LLM?

**Questions:**

Please address the points in the weakness section.

---

> ### Author Response · Authors · 2025-11-28
>
> Q: In line 114, the authors claim that “the assumption is mild and purely about expressivity: any parametric family ….” I understand that the argument is to expand the support by adding an additional element  and extending the family via the construction of . However, to establish powerfulness, shouldn’t the proof operate within this expanded family as well? This reasoning works for exponential families, but I do not see how it holds for arbitrary parametric families.
>
> A: Thank you for this helpful comment. The phrase “any parametric family” in line 114 is a misstatement and should be restricted to exponential families, in the sense made precise in our “powerful family” section. Our intention is not to take an arbitrary model class and enlarge it ad hoc, but rather to work from the outset within an exponential-family framework, where an appropriate choice of sufficient statistics and a rich enough parameter region yields a family expressive enough to satisfy the powerfulness condition. In the revision we have (i) replaced “any parametric family” by “any exponential family”.
>
> Q: It seems to me that the proposed algorithmic approach is specifically tailored to exponential families. Can the method can be extended to more general model classes?
>
> A: Our focus on exponential families is motivated by the kinds of models we have in mind: standard autoregressive LMs (including many LLMs) parameterize next-token distributions via a softmax over logits, which is exactly a categorical exponential family, and the induced sequence distributions can be viewed as (curved) exponential-family models or closely approximated by such models on the parameter ranges of interest. Exponential families are known to form rich approximation classes (e.g., sequences of finite- or infinite-dimensional exponential families can approximate large classes of distributions in KL), so for more general model classes our framework can be applied by working with suitable exponential-family approximations of their sequence distributions. Extending the analysis to make this approximation step explicit and quantify the resulting error is an interesting direction for future work.
>
> Q: Given that the paper’s main contribution is computational efficiency, it would be helpful to include some empirical validation, which shouldn't be too hard.
>
> A: Thank you for the suggestion. In the revision, we have added a new section with a controlled synthetic task in which we run our algorithm in the same oracle setting as in the theory.
>
> Q: Is the sentence in line 200 generated by LLM?
>
> A: Thank you for catching this. The sentence in line 200 is an editing artifact that was accidentally left in the source when we were revising the text; it is not part of the intended exposition, and it has been removed.

---

### Official Review · Reviewer_eVZL · 2025-10-30

**Soundness:** 3
**Presentation:** 3
**Contribution:** 2
**Rating:** 4
**Confidence:** 3

**Summary:**

This paper presents a novel and theoretically-grounded algorithm to mitigate hallucinations in generative models. The authors formulate the problem as a constrained optimization task: maximizing the likelihood of observed valid data while ensuring the model's hallucination rate (the probability mass assigned to a set of invalid outputs) remains below a predefined threshold `α`. The core contribution is a novel connection between this problem and the field of truncated statistics. By using a Lagrangian relaxation, the authors show that the constrained objective can be related to the negative log-likelihood of a truncated distribution, which is a well-studied convex problem for exponential families.

The paper proves that for "powerful" model families (a class they define, which includes exponential families), this approach leads to a computationally and query-efficient algorithm. The proposed method, based on projected stochastic gradient descent (PSGD) over sublevel sets of the loss function, is provably effective and settles an open problem posed by Hanneke et al. (2018) regarding the existence of such an algorithm.

**Strengths:**

1. In a field often dominated by empirical results, this work provides rigorous, provable guarantees on computational efficiency, query efficiency, and correctness (i.e., achieving the target hallucination rate). Proving that the problem is tractable under the "powerful model" assumption, while being NP-hard in general, is a solid theoretical contribution.
2. A key strength is the paper's novel and principled formulation. By framing hallucination reduction as a constrained optimization problem and connecting it to truncated statistics, the authors leverage established mathematical tools to develop a rigorous solution.

**Weaknesses:**

1. The theoretical guarantees hinge on the "powerful model" assumption, primarily exemplified by exponential families. This creates a significant gap, as it is unclear whether complex Transformer architectures satisfy this property, limiting the direct applicability of the theory to state-of-the-art models.
2. The work is entirely theoretical and lacks empirical experiments, even on synthetic data. This absence makes it difficult to assess the algorithm's practical performance and whether its polynomial complexity is feasible for large-scale applications.
3. The framework presumes access to a perfect and cost-free "invalidity oracle." This overlooks the practical challenges of implementing such an oracle, which is often a noisy, biased, and expensive component (e.g., human feedback or another model) in real-world systems.
4. The algorithm's multi-level structure, involving nested optimization loops and projections, appears complex to implement and tune. The practical difficulties of managing these components, particularly the projection onto sublevel sets, may hinder its adoption compared to simpler methods.

**Questions:**

Refer to Weaknesses

---

> ### Author Response · Authors · 2025-11-28
>
> We answer on all the questions posed on the weakness in order.
>
> 1). Thank you for raising this point, which we believe is important to address. Our intention is not that the “powerful model” assumption is a strong structural restriction, but that it captures a mild capacity requirement: the family should be expressive enough. On a finite outcome space, any parametric model can be embedded into the full categorical (multinomial) exponential family, where it appears as a smooth submanifold (a curved exponential family). Moreover, there are classical approximation results showing that exponential families (including sequences of finite-dimensional ones and infinite-dimensional variants) can approximate smooth densities arbitrarily well on compact parameter regions (Barron & Sheu, 1991; Sriperumbudur et al., 2017). In this sense, working with exponential-family surrogates is a standard modeling device and not very restrictive.
>
> In the Transformer architecture, the conditional next-token distribution is obtained by applying a learned linear map followed by a softmax to the logits, producing a categorical distribution over the vocabulary (Bengio et al., 2003; Vaswani et al., 2017). The categorical/multinomial distribution is the canonical exponential-family model on a finite set with softmax, so at the token level the model is exactly a categorical exponential family. At the level of full outputs (e.g., stopped sequences), this induces a regular parametric family over a finite sequence space, which can be viewed as a curved exponential family inside the full categorical exponential family (Amari & Nagaoka, 2000). On the parameter areas of interest, such curved models can be approximated arbitrarily well by suitable finite-dimensional exponential families (Barron & Sheu, 1991; Sriperumbudur et al., 2017).
>
> 2). We agree that a concrete experiment would help clarify how our algorithm behaves in practice, and we have added such an example to the revised manuscript (see the new “Gaussian Example” subsection). In the revision, we consider a one-dimensional Gaussian model and construct a synthetic dataset of valid examples supported on a highly nontrivial validity set $S$: a half-line together with a collection of small “islands” around a few negative points. The complement $S^c$ plays the role of the hallucination set. By design, the geometry of $S$ is disconnected and irregular, so a learner that only sees valid samples and only performs unconstrained maximum likelihood has no way to reconstruct $S$ or control the probability mass assigned to $S^c$.
>
> Within this Gaussian exponential family, we compare (i) standard maximum-likelihood training on all valid samples with (ii) our constrained projected SGD procedure. As described in the manuscript, unconstrained maximum likelihood produces a broad Gaussian that fits the valid data reasonably well but assigns substantial probability to the hallucination region. In contrast, running our algorithm with a target hallucination rate $ \alpha $ yields a Gaussian that is shifted into the valid region and more concentrated around the observed examples; its probability mass on $S^c$ is driven below the prescribed threshold while the loss on the valid data improves. This example concretely illustrates the central message of the paper: the constrained objective behaves differently from standard likelihood, actively trading off fit and validity in the way our analysis predicts.
>
> 3). We agree that in practice invalidity feedback is noisy, biased, and expensive, and we do not claim otherwise, however, despite their cost, real systems already make heavy use of exactly this kind of feedback signal: human preference labels, red-teaming, and model-based filters are all practical instantiations of an “invalidity oracle”.
>
> 4). We agree that the way the algorithm is presented—with an outer loop over sublevel thresholds and inner projected SGD— makes it look more complicated than what one would actually implement. In the analysis, the loop over sublevel sets is used to guarantee that we operate in a region where the hallucination mass does overshoot; this is for the technical analysis in terms of achieving optimality.
>
> In practice, one can avoid this nested structure. A simple implementation is to fix a single sublevel set of the form $ \mathcal{L}(\theta) \le L_{\min} + C \log(1/\alpha) $
> for a reasonable constant \(C\), and run a single PSGD run constrained to this region. The order of $\log(1/\alpha)$ guarantees that any minimizer of interest lies in the domain. If needed, one can perform a one-dimensional search over $C$ (in a few trials) rather than looping over many sublevels, so the “projection onto sublevel sets” used in the proof reduces in practice to checking whether $\(\mathcal{L}(\theta)\)$ stays below a chosen threshold and adjusting that threshold via a simple linear search.

---

### Official Review · Reviewer_d5VF · 2025-10-31

**Soundness:** 3
**Presentation:** 3
**Contribution:** 3
**Rating:** 6
**Confidence:** 3

**Summary:**

This paper formulates hallucination reduction as likelihood maximization under an explicit upper bound on the probability of invalid generations. It connects this constrained objective to truncated statistics and proposes a projected SGD–based procedure that actively queries invalid outputs, operating over expressive exponential families. The authors prove computational/query efficiency and provide conditions (“powerful” models) under which proper learning achieves a target hallucination rate.

**Strengths:**

1) The work gives a statistically, query-, and computationally efficient proper learner that drives the hallucination rate to any target \alpha addressing the efficiency question posed by Hanneke et al. (2018).

2) A key theoretical contribution is the novel bridge established between the generative model hallucination problem and the field of truncated statistics. By reformulating the constrained optimization via a Lagrangian relaxation, the objective function can be mapped to a truncated negative log-likelihood, making the problem tractable and solvable with convex optimization techniques for certain model families.

3) The authors develop a concrete algorithm that is proven to work for "powerful" exponential families, a class of models shown to be sufficiently expressive for this task. The algorithm is efficient, requiring only a polynomial number of invalidity queries and computation time to find a solution that meets the desired hallucination rate while remaining near-optimal in data likelihood.

**Weaknesses:**

The paper is entirely theoretical and does not include any experiments to validate its claims. While the theoretical contributions are strong, the work would be significantly strengthened by empirical results, even on a synthetic toy model. Such experiments could demonstrate the algorithm's practical behavior, verify that it achieves the target hallucination rate, and provide insight into its performance in a controlled setting.

**Questions:**

1) The paper's theoretical results are derived for exponential families, which creates a gap between this setting and the large-scale models like LLMs that motivate the work. Could the authors add a discussion on the primary challenges or potential paths for extending this framework to more general architectures, such as Transformers?

2) Could you report the empirical results on some widely-used hallucination benchmark datasets to validate the proposed method?

---

> ### Author Response · Authors · 2025-11-28
>
> Q: The paper's theoretical results are derived for exponential families, which creates a gap between this setting and the large-scale models like LLMs that motivate the work. Could the authors add a discussion on the primary challenges or potential paths for extending this framework to more general architectures, such as Transformers?}
>
> A: Our exponential-family setting is not meant to exclude Transformers. In standard autoregressive LMs, each step applies a softmax to logits, i.e., a categorical exponential family, and the product of these conditionals induces a regular parametric (curved exponential-family) model on sentences. In our analysis we phrase everything in terms of an abstract exponential family over the sentence space, but the algorithm itself only requires sampling from the model together with oracle access to a set (S) of hallucinating sentences (i.e., the ability to test whether a sampled sentence is hallucinating), which a Transformer LM can provide. The remaining challenge, which we view as an important direction for future work, is to design efficient, online implementations of our framework that operate in this autoregressive, sample-and-oracle setting without ever computing full sequence probabilities or summing over the entire sentence space. We will add this discussion in the revised manuscript.
>
> Q: Could you report the empirical results on some widely-used hallucination benchmark datasets to validate the proposed method?
>
> A: In the revision, we have added a new section with a controlled synthetic “hallucination” task: data are observed only on a valid region, and outputs outside this region are treated as hallucinations. This experiment illustrates how our method affects the probability of hallucinations in a setting that matches the assumptions of our analysis. A systematic study on large benchmark datasets would require a more elaborate implementation and is beyond the scope of the present work.

---

### Author Response · Authors · 2025-11-28
**General Rebuttal**

We really appreciate that the reviewers find our paper novel and timely, and we are grateful for their thoughtful feedback, which has helped us improve both the presentation and scope of the work. In particular, we are encouraged that Reviewer d5VF confirms that our statistically, query-, and computationally efficient proper learner resolves the open question of Hanneke et al. (2018); that Reviewer eVZL emphasizes the importance of providing rigorous guarantees in a largely empirical area, and notes that our paper constitutes a solid mathematical foundations contribution in the area; and that Reviewer T5Vg finds our contribution both interesting and significant.

The main concerns focused on (i) whether our exponential-family (“powerful”) framework could be restrictive and whether it can capture the complexity of modern architectures, and (ii) the lack of empirical validation. In the revised manuscript, we (a) add a new section with a controlled synthetic “hallucination” task, showing how access to an invalidity oracle affects learning in exactly the oracle setting of our theory, and (b) expand the introduction around our main theorem to explain why the exponential-family framework is well aligned with practice: standard autoregressive models use softmax-based categorical parameterizations (see, e.g., Bengio et al., 2003; Vaswani et al., 2017; Yang et al., 2018). In addition, exponential families are known to have strong approximation properties (e.g., Barron and Sheu, 1991; Amari and Nagaoka, 2000; Sriperumbudur et al., 2017), so our assumptions should be viewed as expressive rather than restrictive.

References:
- Bengio, Y., Ducharme, R., Vincent, P., and Jauvin, C. (2003). “A Neural Probabilistic Language Model.” Journal of Machine Learning Research.
- Vaswani, A., Shazeer, N., Parmar, N., Uszkoreit, J., Jones, L., Gomez, A. N., Kaiser, Ł., and Polosukhin, I. (2017). “Attention Is All You Need.” NeurIPS.
- Yang, Z., Dai, Z., Salakhutdinov, R., and Cohen, W. W. (2018). “Breaking the Softmax Bottleneck: A High-Rank RNN Language Model.” ICLR.
- Barron, A. R., and Sheu, C.-H. (1991). “Approximation of Density Functions by Sequences of Exponential Families.” Annals of Statistics.
- Amari, S., and Nagaoka, H. (2000). Methods of Information Geometry. American Mathematical Society.
- Sriperumbudur, B. K., Fukumizu, K., Gretton, A., Hyvärinen, A., and Kumar, R. (2017). “Density Estimation in Infinite Dimensional Exponential Families.” Journal of Machine Learning Research.

---

### Meta-Review · Area_Chair_EksU · 2026-01-09

**Summary:**

The paper proposes an algorithm to reduce hallucinations in generative models by posing the problem within the framework of constrained optimization. The work proves theoretical efficiency of the method and how it can achieve a target hallucination rate under suitable assumptions.

Reviewers generally appreciated the theoretical contributions, but all raised concerns about a lack of experiments to complement the theoretical results and practical implementability of the method. Synthetic experiments to supplement the theory would make this a stronger work. These have not been fully addressed in the rebuttal, and it remains unclear how large the gap between the theoretical results and modern deep learning practice is and whether it can be bridged.

Therefore, the paper is not recommended for acceptance in its current form.

**Reviewer Concerns:**

Reviewer's concerns about limitations of exponential family assumptions and expressivity of the approximation families (eVZL, T5Vg, d5VF) have been addressed in the rebuttal.

All reviewers raised concerns about lack of experiments. The rebuttal adds a simple one-dimensional Gaussian example to the manuscript, but reviewers were likely expecting slightly larger numerical experiments and implementing the proposed algorithm to run on a computer.

Reviewer eVZL raises concerns on whether the proposed algorithm can be implemented in practice, and while the rebuttal provides some practical simplifications, concerns about efficiency of computing the projections remain open.

**Reviewer Scores:**

Reviewers d5VF, eVZL, T5Vg gave ratings of 6, 4, 6. d5VF, eVZL and T5Vg all likely would have maintained their score as the rebuttal did not fully address the lack of experiments.

---

### Decision · Program_Chairs · 2026-01-26

Reject